# Multisensory gaze stabilization in response to subchronic alteration of vestibular type I hair cells

**Louise Schenberg[1], Aïda Palou[2,3,4], François Simon[1,5], Tess Bonnard[1], Charles-Elliot Barton[1], Desdemona Fricker[1], Michele Tagliabue[1], Jordi Llorens[2,3,4]\*, Mathieu Beraneck[1]\***

[1]Université Paris Cité, CNRS UMR 8002, INCC - Integrative Neuroscience and Cognition Center, Paris, France; [2]Departament de Ciències Fisiològiques, Universitat de Barcelona, Barcelona, Spain; [3]Institut de Neurociènces, Universitat de Barcelona, Barcelona, Spain; [4]Institut d'Investigació Biomèdica de Bellvitge (IDIBELL), l'Hospitalet de Llobregat, Spain; [5]Department of Paediatric Otolaryngology, Hôpital Necker-Enfants Malades, Paris, France

**\*For correspondence:**
jllorens@ub.edu (JL);
mathieu.beraneck@cnrs.fr (MB)

**Competing interest:** The authors declare that no competing interests exist.

**Abstract** The functional complementarity of the vestibulo-ocular reflex (VOR) and optokinetic reflex (OKR) allows for optimal combined gaze stabilization responses (CGR) in light. While sensory substitution has been reported following complete vestibular loss, the capacity of the central vestibular system to compensate for partial peripheral vestibular loss remains to be determined. Here, we first demonstrate the efficacy of a 6-week subchronic ototoxic protocol in inducing transient and partial vestibular loss which equally affects the canal- and otolith-dependent VORs. Immunostaining of hair cells in the vestibular sensory epithelia revealed that organ-specific alteration of type I, but not type II, hair cells correlates with functional impairments. The decrease in VOR performance is paralleled with an increase in the gain of the OKR occurring in a specific range of frequencies where VOR normally dominates gaze stabilization, compatible with a sensory substitution process. Comparison of unimodal OKR or VOR versus bimodal CGR revealed that visuo-vestibular interactions remain reduced despite a significant recovery in the VOR. Modeling and sweep-based analysis revealed that the differential capacity to optimally combine OKR and VOR correlates with the reproducibility of the VOR responses. Overall, these results shed light on the multisensory reweighting occurring in pathologies with fluctuating peripheral vestibular malfunction.

## eLife assessment

This paper provides a **fundamental** expansion of vestibular compensation into transient and partial dysfunction, as well as insights into the adaptation of visual reflexes in this process. The conclusions are **convincingly** supported with paired histological and behavioral measurements, which are additionally modeled for further interpretation. This work would be of interest to neuroscientists working in multisensory integration and recovery mechanisms.

## Introduction

The vestibular system is well-preserved amongst vertebrates, participating in essential functions such as balance, postural control and, together with the optokinetic system, gaze stabilization (*Straka et al., 2016*; *Wibble et al., 2022*). Beyond these recognized roles, vestibular signals also contribute to cognitive processes for example spatial orientation and navigation (*Cullen, 2019*), or body

representation (*Lopez et al., 2012*; *Facchini et al., 2021*). Because of its involvement in many basic functions important in our daily life, vestibular pathologies affecting the inner ear are associated with a significant deterioration of the well-being of patients (*Möhwald et al., 2020*) and represent an important public health concern (*Agrawal et al., 2009*; *Agrawal et al., 2013*).

Research on post-lesion plasticity following permanent vestibular loss has shed light on the neural plastic mechanisms that follow a chronic unilateral or bilateral vestibular lesion, a process referred as "vestibular compensation" (*Brandt et al., 1997*; *Cullen et al., 2010*; *Beraneck and Idoux, 2012*). The compensation taking place after the lesion is known to involve dynamical multisensory reweighting of proprioceptive and visual inputs and of internal efferent copies (*Cullen et al., 2010*; *Sadeghi et al., 2012*; *Sadeghi and Beraneck, 2020*). While total and permanent lesions offer the experimental opportunity to characterize drastic cellular and molecular changes triggered by the total silencing of the vestibular endorgans, they imperfectly mimic clinical situations where peripheral vestibular function loss is only partial and/or transient (*Bisdorff et al., 2009*; *Lopez-Escamez et al., 2015*; *Brandt and Dieterich, 2017*). To better model fluctuating inner ear function, protocols based on subchronic exposure to an ototoxic substance, 3,3'-iminodiproprionitrile (IDPN) were first introduced in the rat (*Seoane et al., 2001*; *Sedó-Cabezón et al., 2014*; *Sedó-Cabezón et al., 2015*; *Martins-Lopes et al., 2019*) and more recently in the mouse (*Greguske et al., 2019*). Subchronic exposure to IDPN in drinking water at low doses allowed for progressive ototoxicity, leading to a partial and largely reversible loss of function. Although the subchronic IDPN protocol was shown to cause postural and locomotor deficits (*Martins-Lopes et al., 2019*), its effects on the gaze stabilizing reflexes, namely the vestibulo-ocular and optokinetic reflexes, have not yet been described.

The primary objective of the present study is first to assess how subchronic exposure to IDPN may affect the function of the different vestibular endorgans. To that end, we took advantage of our recently described methodology (*Simon et al., 2020*; *Simon et al., 2021*) using canal-specific and otolith-specific tests. Quantification of the vestibulo-ocular reflexes is a sensitive and specific method to assess the functionality of the sensory-motor vestibular pathway; it is the most used test in clinics, and highly correlates to quality-of-life reports in patients suffering from acute peripheral diseases (*Möhwald et al., 2020*).

The secondary objective is to determine whether visual substitution occurs following transient and partial vestibular loss. Optogenetic stimulation of the vestibular pathway demonstrated the recruitment of circuits involved in visual processing at the midbrain, thalamic and cortical regions (*Leong et al., 2019*). Recent imaging studies in rodents have shown that acute vestibular loss triggers brain-wide adaptive plasticity in circuits known to be involved in visual processing (*Zwergal et al., 2016*; *Grosch et al., 2021*). In addition, it was previously shown that OKR plasticity is triggered during vestibular compensation following a permanent vestibular lesion (*Faulstich et al., 2006*; *Nelson et al., 2017*).

We report that 6 weeks of IDPN subchronic treatment affects both the canal- and otolith-dependent vestibulo-ocular reflexes and that organ-specific loss of type I hair cells (HC) correlates with individual mice's impairments. We show that optokinetic adaptive compensation is frequency-specific and delayed with respect to the VOR changes. OKR changes occur at the frequencies where physiologically the vestibular inputs dominate visuo-vestibular gaze stabilization. We demonstrate that despite the significant recovery of their vestibulo-ocular reflexes, the visuo-vestibular integration remains notably impaired in some IDPN-treated mice. We suggest that the 'noisiness' of the recovered vestibular signal affects their capacity to optimally combine visual and vestibular responses. Overall, these results shed light on the dynamic of multisensory reweighting in patients suffering from fluctuating peripheral vestibular malfunction.

## Results
### Effects of the subchronic treatment of IDPN on the canal- and otolith-dependent VOR

To investigate the effects of the IDPN on vestibulo-ocular reflexes (VOR), animals were exposed to the ototoxic compound in the drinking water for six weeks (Treatment period), followed by 6 weeks of standard drinking water without IDPN (Washout period). The VOR were quantified every two weeks using canal-specific and otolith-specific tests. Horizontal sinusoidal rotations in the dark were performed

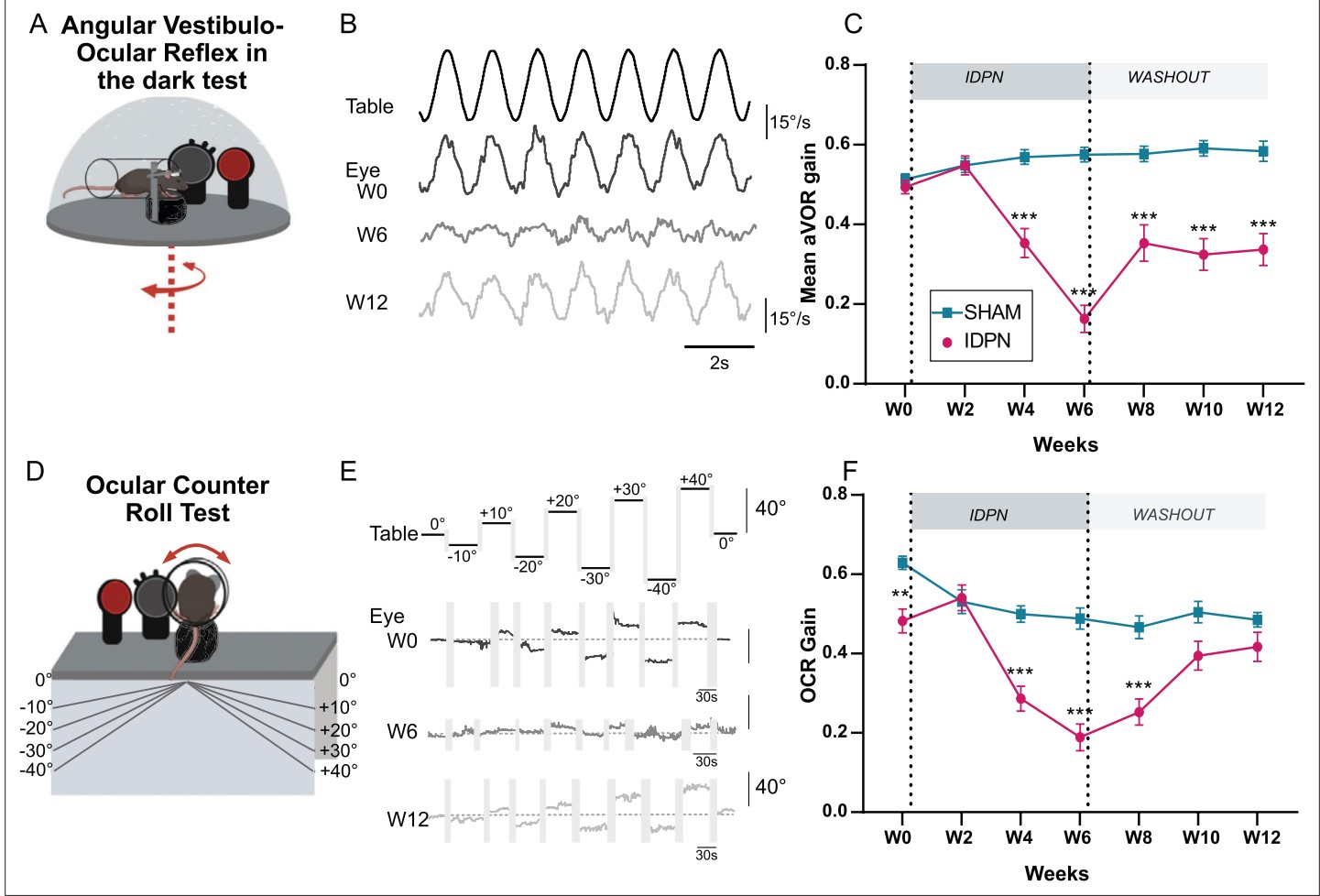

**Figure 1.** Effects of subchronic IDPN on canal- and otolithic- dependent VOR. (**A**) Illustration of the angular horizontal vestibulo-ocular reflex (aVOR) set-up. All tests are performed in complete dark. (**B**) Examples of velocity aVOR traces in response to table rotations (1 Hz at 30°/s) recorded in the dark in an IDPN mouse before (W0, corresponding gain of 0.785; VAF of 0.98), after 6 weeks of treatment (W6, gain: 0.14; VAF: 0.61) and 6 weeks of washout (W12, gain:0.48; VAF: 0.92). Right movement is represented up. (**C**) Mean aVOR gain of SHAM (n=22) and IDPN (n=21) mice during the protocol (repeated measures ANOVA). (**D**) Illustration of the ocular- counter roll (OCR) set-up. (**E**) Examples of raw OCR traces at W0, W6 and W12 in a IDPN mouse recorded in the dark. Tilt to the right is represented up (positive values). (Left) eye elevation is represented up. (**F**) Mean OCR gain of SHAM (n=14) and IDPN (n=13) mice (mix-model ANOVA). We note that there was a significant difference between SHAM and IDPN during the initial measurements at W0. However, at this time point mice were not yet separated into different groups. This incidental difference completely disappeared on the measurement performed at W2. (*p<0.05; **p<0.01; ***p<0.001). Error bars represent ± SEM.

The online version of this article includes the following source data and figure supplement(s) for figure 1:

**Source data 1.** Effects of subchronic IDPN on canal- and otolithic- dependent VOR.

**Figure supplement 1.** Effects of subchronic IDPN on canalar- and otolith-dependent VOR.

and oculomotor responses were recorded using video-oculography (*Figure 1A*) to study the impact of IDPN treatment on the canal-dependent angular VOR (aVOR). Typical raw aVOR traces are shown in *Figure 1B*. At week 6 (W6), the amplitude of the eye movements was distinctly reduced compared to W0, while at W12 the amplitude of the response appeared partially restored. The dynamics of decrease and recovery of the mean aVOR gain over the course of the protocol are reported in *Figure 1C* for both IDPN (n=21) and SHAM (n=22) groups. The evolution over time of the aVOR gain is significantly different for the two groups of mice (ANOVA Weeks x Group interaction, $F_{(6,246)}=29,949$, $p<10^{-4}$). Before treatment, both groups responded similarly to the sinusoidal stimulations and their aVOR gain remained unchanged through 2 weeks of treatment (W2). However, starting W4 the aVOR gain of IDPN group significantly decreased with respect to W0 (Newman-Keuls post hoc test: IDPN W0 vs W4 $p<10^{-4}$, see *Table 1*) and was significantly lower compared with the SHAM group (W4, IDPN vs

**Table 1.** Statistics table of the aVOR gain for the IDPN-treated group.

| W0 | W2 | W4 | W6 | W8 | W10 | W12 | |
|---|---|---|---|---|---|---|---|
|  | ns | *** | *** | *** | *** | *** | W0 |
| ns |  | *** | *** | *** | *** | *** | W2 |
| *** | *** |  | *** | *** | ns | ns | W4 |
| *** | *** | *** |  | * | *** | *** | W6 |
| *** | *** | *** | * |  | *** | *** | W8 |
| *** | *** | ns | *** | *** |  | ns | W10 |
| *** | *** | ns | *** | *** | ns |  | W12 |

SHAM p<$10^{-4}$). The aVOR gain of the IDPN group remained lower compared to SHAM through the rest of the protocol (W6 IDPN vs SHAM p<$10^{-4}$., W8 IDPN vs SHAM p<$10^{-4}$, W10 IDPN vs SHAM p<$10^{-4}$, W12 IDPN vs SHAM p<$10^{-4}$) with a minimum reached at W6 corresponding to ~2/3 of aVOR loss (IDPN W4 vs W6 p<$10^{-4}$, IDPN W6 vs W8 p=0.043762). At the end of the 6 weeks of washout, the mean aVOR significantly improved (IDPN W6 vs W12 p<$10^{-4}$) to levels observed at W4 (IDPN W4 vs W12 p=0.561969). Notably, the amplitude and dynamic of gain changes were similar for all frequencies >0.2 Hz (see *Figure 1— figure supplement 1A*). However, at the lowest frequency tested (0.2 Hz), aVOR gain decrease reached significance only at W6.

aVOR responses were further modified by significant phase leads that affected all frequencies starting W6 (*Figure 1—figure supplement 1B*, ANOVA repeated measures Weeks x Group Interactions F(6, 246)=14.528, p<$10^{-4}$). Overall, canalar responses remained unaffected until week 2, but the amplitude and/or timing of the aVOR was abnormal from week 4 until the end of the protocol, despite a significant recovery of angular VOR responses observed during the washout period.

To determine whether otolith-dependent VOR was also affected by IDPN treatment, ocular-counter roll responses (OCR) were tested during static lateral inclination in the range ± 40° (OCR; *Figure 1D*). Examples of raw traces and quantification of the response are shown in *Figure 1E*, and mean gain of the OCR of each group is plotted in *Figure 1F* (n=13 IDPN, n=14 SHAM). The modulations of responses amplitude were significantly different between IDPN and SHAM groups (ANOVA repeated measures Weeks x Group Interaction F(6, 150)=7.7411 p<$10^{-4}$). We note that there was a significant difference between SHAM and IDPN during the initial measurements at W0, before any treatment. However, at this timepoint mice were not yet separated into different groups. This incidental difference completely disappeared on the measurement performed at W2 (W2 IDPN vs SHAM p=0.8135). While the gain of the SHAM group stays in a 0.5–0.6 range over the whole duration of the protocol, the responses of the IDPN group significantly decreased at W4 compared to the SHAM group (W4 IDPN vs SHAM p<$10^{-4}$). This decrease was larger at W6 (IDPN W4 vs W6 p=0.023) and stayed significantly different from the SHAM group until W8. At W10 and W12, the OCR of IDPN group recovered to a level comparable to the SHAM group (W10 IDPN vs SHAM p=0.1262, W12 IPDN vs SHAM p=0.3385).

Sub-chronic treatment of IDPN was also investigated through the dynamic Off Vertical Axis Rotation (OVAR) test (*Figure 1—figure supplement 1C*), which primarily reflects maculo-ocular (dominantly otolithic) responses integrated by central vestibular pathways. The maculo-ocular reflex (MOR) bias decreased significantly compared to the SHAM group (ANOVA Weeks x Group, F(6, 132)=10.076, p<$10^{-4}$) starting W2 of treatment (W2 IDPN vs SHAM p=0.0018), demonstrating that ototoxicity already affected the vestibular system at this early time point. The maximal decrease was reached at W6 (W6 IDPN vs SHAM p=0.00012) and recovery led to normal responses at W12 (W12 IDPN vs SHAM p=0.4709).

## Comparison of otolith- and canal-dependent plasticity in individuals

To compare the dynamic of canal- and otolith-dependent VOR alterations, the paired VOR and OCR gains measured in individuals from the IDPN group (n=13) are plotted together (*Figure 2*). Alteration in canal- and otolith-dependent responses followed a very similar time course (OCR vs aVOR at 1 HZ on *Figure 2A*; similar patterns were obtained with the other tested frequencies (not shown)).

To investigate further the organ-specific responses, the individual values of VOR gain at 1 Hz (left panel) and OCR slope (right panel) are plotted in *Figure 2B*. These plots show the variability of the responses observed in different individuals. 2/3 of the mice had aVOR responses lowered by more than 50%, while 1/3 had milder aVOR impairments. In most cases, however, the aVOR gain decreased notably between W4 and W6 and started to recover from W8 (compare individual slopes

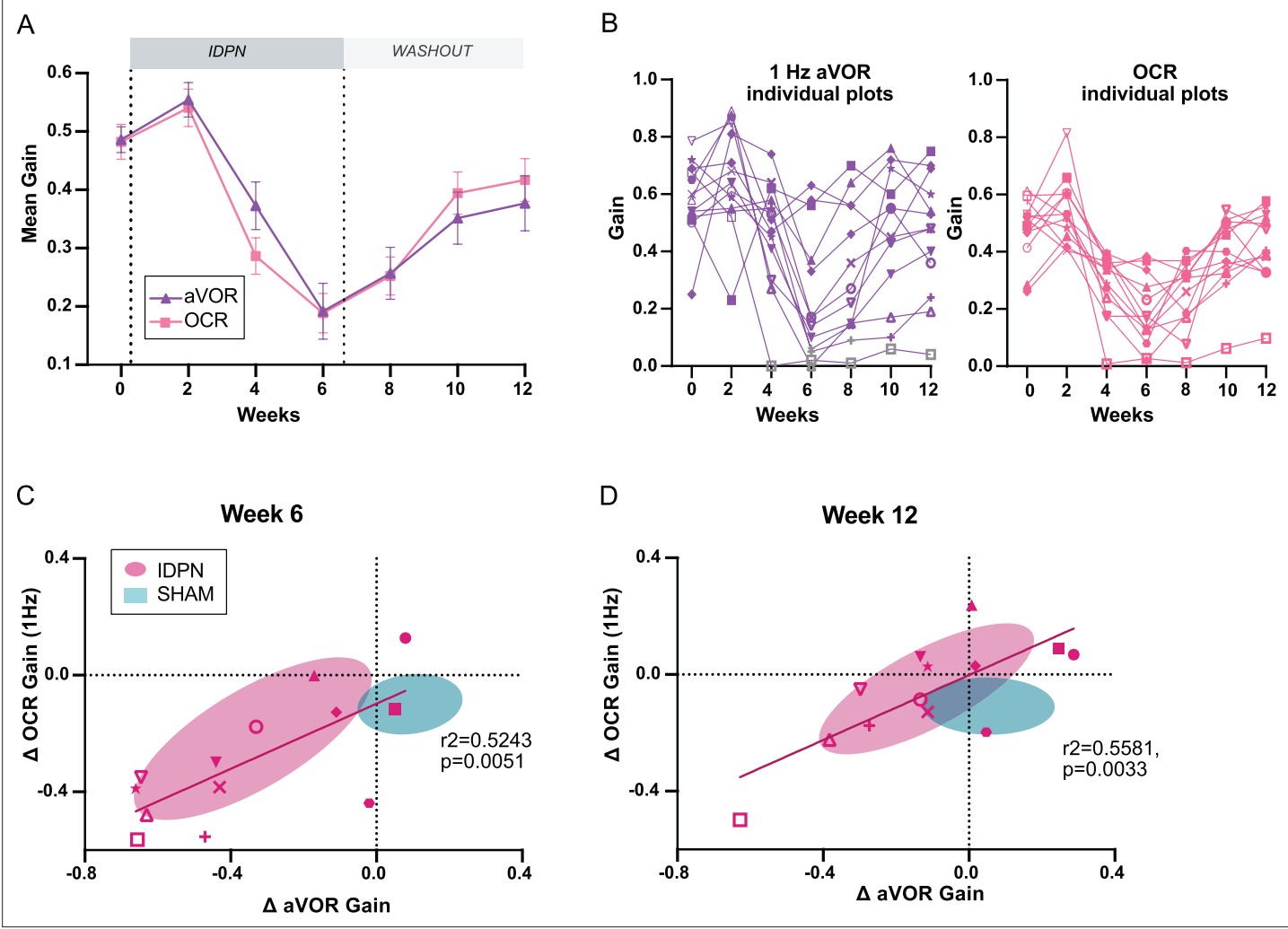

**Figure 2.** Comparison of the dynamics of canalar and otolithic loss of function. (**A**) Evolution of the Mean Gain of the aVOR (1 Hz) and OCR responses of IDPN mice (n=13) during the treatment and washout periods. (**B**) Individual gains for 1 Hz aVOR (left) and OCR (right) of the 13 IDPN mice, identified with similar symbols. The grey symbols in the left panel correspond to aVOR gain values associated with VAF <0.5. (**C, D**) Individual ΔaVOR gains as a function of individual ΔOCR gains at W6 (**C**) and W12 (**D**) compared to W0, for SHAM (n=14) and IDPN (n=13) mice. The linear regression corresponds to IDPN values is represented, as well as the 50% confidence interval of each group (shaded areas). The symbols for each animal are the same in panels B, C, and D. (*p<0.05; **p<0.01; ***p<0.001). Error bars represent ± SEM.

The online version of this article includes the following source data for figure 2:

**Source data 1.** Comparison of the dynamics of canalar and otolithic loss of function.

in *Figure 2B*). Similarly, OCR gains were variable between individuals. However, the dynamic of the variations in otolith-dependent responses followed a pattern comparable to that of the canal-dependent changes.

To determine whether the changes in canal- and otolith-dependent responses were proportional, individual variations in gain since pre-treatment of the aVOR and OCR were compared at W6 (*Figure 2C*) and W12 (*Figure 2D*). At W6, there was a significant correlation between the amplitude of the changes observed in aVOR and OCR, such that canalar and otolithic loss of function was proportional (slope of regression line: 0.5586). Despite the general recovery, this significant correlation was preserved at W12: the recovery in canal-dependent responses was proportional to the recovery in otolith-dependent responses (slope of regression line: 0.5566). Overall, these results suggest that the subchronic IDPN treatment similarly affects the vestibulo-ocular reflexes that depend on semi-circular canals and on the otolithic organs, respectively.

## Effects of the IDPN exposure on the number of hair cells in the vestibular epithelia

To correlate the vestibular loss of function to the structural changes induced by the ototoxic compound, vestibular epithelia were dissected and labelled to assess the number of hair cells (HC) in the organs. One horizontal semi-circular canal and one utricule were harvested at W6 for n=7 IDPN and n=4 SHAM, and at W12 for n=8 IDPN and n=4 SHAM, and each organ epithelium was divided into the central and peripheral region to differentiate the possible participation of zone- and organ-specific HC to the vestibular function. Vestibular HC were labelled with type-specific immunomarkers in both endorgans (*Figure 3*). Confocal immunostaining data are presented in *Figure 3A1* for the horizontal semi-circular canals and *Figure 3C1* for the utricule. Six-week long treatment of IDPN exposure led to a significant reduction in the labelling of type I HC in the canal crista (quantified in *Figure 3A2*, Non parametric Kruskal-Wallis test, SHAM vs IDPN Spp1, p=0.0322; SHAM vs IDPN CASPR1, p=0.0268; *Figure 3—figure supplement 1A1 and B1* for the peripheral zones) and in the otolith macula (*Figure 3C2*, Non parametric Kruskal Wallis test SHAM vs IDPN Spp1, p=0.0279; SHAM vs IDPN CASPR1, p=0.0343). However, the treatment did not alter type II HC-specific labelling as the number of Calre + cells in the IDPN group was similar to the SHAM in all regions of either vestibular endorgans (*Figure 3A2*, canal crista, Calre: Non parametric Kruskal-Wallis test IDPN vs SHAM, p>0.9999; *Figure 3C2*, otolith macula, Calre: SHAM vs IDPN, *P*>0.9999). Importantly, the total number of HC marked with Myo7a (labelling both type I and type II HC) was not significantly different between the IDPN-treated mice and the SHAM, suggesting that the alteration of type I HC is not associated with cell loss. Moreover, at W12 the number of type I HC labelled in the IDPN-treated mice was no longer significantly different from the SHAM group in either endorgans (*Figure 3A3* for crista; *Figure 3C3* for macula). The count of type II-specific markers and of non-specific HC markers was not statistically different between SHAM and IDPN either, so that overall no difference persisted at W12, indicating a structural recovery at the end of the washout period.

To determine how the loss of HC correlates with organ-specific functional tests, the numbers of type I and II HC in the central region of the horizontal semi-circular canals are plotted as a function of the mean aVOR gain for mice at W6 and W12 (*Figure 3B1*). Both markers of type I HC (CASPR1 and Spp1, left panels) were significantly correlated with the aVOR gain (linear regression, Spp1: $r^2$=0.5128, p=0.0001; CASPR1: $r^2$=0.4948, p=0.0002), whereas the rather constant number of type II HC in the ampulla did not correlate with the variation observed in aVOR function. Finally, to determine if the correlation was due to the recovery occurring between W6 and W12, the number of CASPR + cells of IDPN mice counted at W12 are plotted as a function either of the individual aVOR gain at W12 (pink squares) or aVOR gain at W6 (black circles), paired by an arrow (*Figure 3B2*). The recovery of the vestibular function between W6 and W12 induces a shift toward the W12 linear regression, reinforcing the notion that the recovery of function was related to the increase in the number of hair cells with normal expression of biochemical markers.

The numbers of type I HC and II HC found in the striolar region of the utricule are similarly plotted in relation to the OCR gain (linear regression, Spp1: $r^2$=0.6011, p<0.0001; CASPR1: $r^2$=0.6342, p<0.0001, *Figure 3D1*). Again, the otolithic function correlated with the number of type I HC, and not type II HC in the central region. Furthermore, the correlation found between the otolithic function and the number of type I HC related to the increase in the number of hair cells after the recovery period (*Figure 3C2*). A similar correlation between organ-specific function and the number of type I HC was also found in the peripheral regions of these organs (see *Figure 3—figure supplement 1C1, C2* for the SCC, and *Figure 3—figure supplement 1D1, D2* for the utricule).

Taken together, comparison at W6 and W12 of non-specific HC and type I-specific HC markers suggest that the ototoxic effect induces a transient biochemical alteration of type I HC rather than a definitive hair cell degeneration. Overall, IDPN-induced alteration of canal and otolith functions are correlated with the ototoxic effect on type I, and not type II, HC in the central and peripheral zones of vestibular endorgans.

## Effects of the subchronic treatment of IDPN on the optokinetic reflex

To determine the effects of the subchronic treatment of IDPN on the optokinetic reflex (OKR), mice were tested with sinusoidal rotations of a virtual drum (*Figure 4A*). An example of raw traces of the evoked horizontal eye movements is shown in *Figure 4B* for different timepoints (0.5 Hz at 10 deg.s$^{-1}$

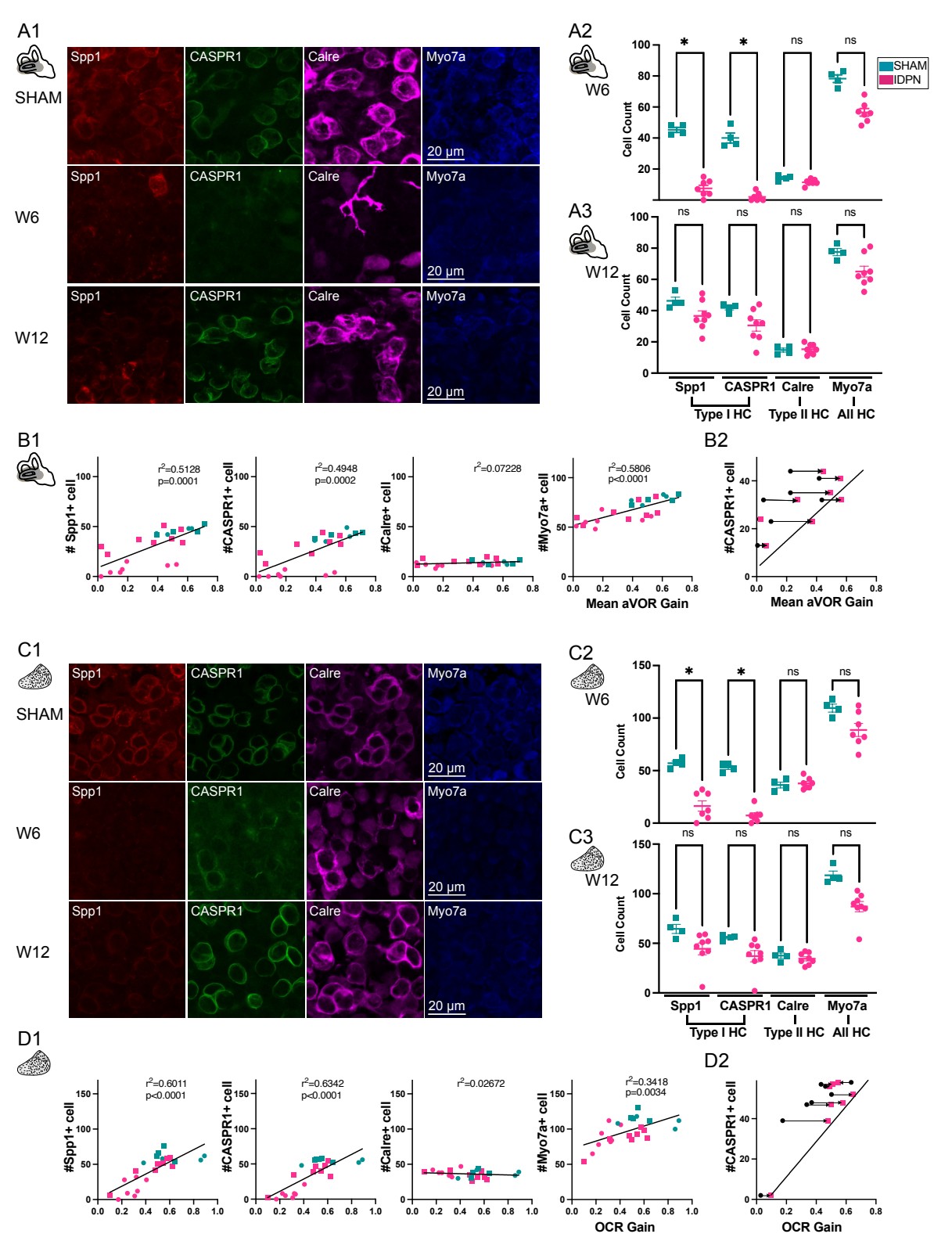

**Figure 3.** Immuno-labelling of HC in the central regions of the horizontal SCC ampulla and striolar region of the utricule Macula. (**A, C**) Immunolabelling of type I HC (Spp1 + and CASPR1+), type II HC (Calre +), or all HC (Myo7a) for the SHAM, IDPN W6 and IDPN W12 groups in the central ampulla of the horizontal canal (**A1**) and central utricular maculae (**C1**). Cell count at W6 and W12 in the central horizontal ampulla (**A2 and A3**) and central utricular maculae (**C2 and C3**) for individual mice (Kruskal Wallis test). (**B1, D1**) Individual number of central Spp1 + type I HC, CASPR1 + type I HC and Calre +

*Figure 3 continued on next page*

*Figure 3 continued*

type II HC, or all HC(Myo7a) as a function of the aVOR gain (**B1**) or OCR gain (**D1**) at W6 (circle) and W12 (squares) groups. The linear regressions correspond to all individuals (n=23 mice). (**B2, D2**) Comparison of the number of CASPR1 type I HC as a function of the aVOR (**B2**) gain or OCR gain (**D2**) at W6 (black and circle) and at W12 (pink and square) for each IDPN mice of the W12 group (n=8). Note that all points are shifted toward the regression line (redrawn from respectively B1 and D1), indicating that the number of cells at W12 better correlates with the recovered aVOR. (*p<0.05; **p<0.01; ***p<0.001). Error bars represent ± SEM.

The online version of this article includes the following source data and figure supplement(s) for figure 3:

**Source data 1.** Immuno-labelling of HC in the central regions of the horizontal SCC ampulla and striolar region of the utricule Macula.

**Figure supplement 1.** Effects of the IDPN on the number of HC in the peripheral regions of the horizontal SCC ampulla and extrastriolar utricule Macula.

optokinetic stimulation). The mean OKR gain for all tested frequencies is plotted in *Figure 4C* for SHAM (n=12) and IDPN-treated mice (n=12). The mean OKR gain of IDPN mice was significantly different from the mean OKR gain of SHAM mice at W8 (interaction between Weeks and Group $F_{(6,132)}=2.9845$, p=0.0091; IDPN W0 vs W8, p=0.0216, see *Table 2* and *Figure 4—figure supplement 1*) where it reached its peak (IDPN W0 vs W8, p=0.00037). The gain stayed significantly larger compared to the SHAM through the end of the washout period (W12 IDPN vs SHAM, p=0.046).

To determine whether this OKR modulation was frequency-specific, the gain of the 5 tested frequencies was compared for W0, W6, W8 and W12 (*Figure 4D*). There was at W0 no difference between SHAM and IDPN mice (*Figure 4D*, left panel, W0). At W6, there was a significant difference between the gain of the IDPN and SHAM groups limited to the frequencies 0.5 and 1 Hz (W6 IDPN vs. SHAM, 0.5 Hz, p=0.013117; W6 IDPN vs. SHAM, 1 Hz, $p<10^{-4}$). At W8, the gains measured at frequencies above 0.2 Hz were significantly increased compared to the SHAM group (W8 IDPN vs. SHAM, 0.33 Hz, p=0.027687; 0.5 Hz, p=0.0025; 1 Hz, p=0.00015). At W12, the gain remained high for both 0.5 and 1 Hz, whereas the gain at 0.33 Hz was no longer significantly larger than the SHAM group (W12 IDPN vs. SHAM, 0.33 Hz, ns; 0.5 Hz, p=0.0069; 1 Hz, p=0.00078). Globally, responses at higher frequencies (0.33, 0.5, and 1 Hz) were significantly increased (ANOVA Weeks x Group x Frequencies interaction, $F_{(24, 528)}=6.5870$, $p<10^{-4}$), whereas responses at the lowest frequencies (0.1 and 0.2 Hz) were not significantly modulated. Additionally, these changes in gain for frequencies >0.2 Hz were not accompanied by changes in the timing (phase) of the OKR (*Figure 4E*; ANOVA, Weeks x Group x Frequencies interaction, p=0.3802).

## Relation between VOR decrease and optokinetic increase

To correlate the changes in vestibular pathway with the frequency-specific changes observed in the optokinetic pathway, we compared the responses in aVOR and OKR obtained at the frequencies that were common between the 2 tests (i.e 0.2 Hz, 0.5 Hz and 1 Hz) for n=12 SHAM and n=12 IDPN treated mice by comparing the increase in the OKR and decrease in VOR relative to W0 (Δmean gain, see *Figure 5*). To determine whether the changes in VOR and OKR were proportional, we compared the paired decrease and increase in the mean ΔaVOR and ΔOKR gain after 8 weeks of treatment, when the difference between the 2 reflexes peaked (*Figure 5A*). The vast majority (n=11/12) of IDPN-treated mice showed a decrease in aVOR and an increase in OKR (top left quadrant corner, *Figure 5B*), which was not the case for SHAM mice (blue squares). However, there was no correlation between the amplitude of the VOR decrease and the amplitude of the OKR increase, that is mice that had the greatest VOR loss did not show the largest OKR increase (slope in the regression line: 0.27, p=0.0834).

To determine whether the parallel changes in OKR and VOR are frequency-specific, *Figure 5C* compares the gains measured at W0 and W8 for each frequency. At 0.2 Hz, there was a significant decrease in aVOR (IDPN W0 vs. W8 0.2 Hz $p<10^{-4}$); however, the OKR was not modified. On the other hand, at 0.5 and 1 Hz significant VOR decreases (IDPN W0 vs. W8, 0.5 Hz, $p<10^{-4}$; 1 Hz, $p<10^{-4}$) were accompanied by a significant OKR increase (IDPN W0 vs. W8, 0.5 Hz, $p<10^{-4}$; 1 Hz, $p<10^{-4}$). To examine this frequency-selective increase, the percentage of change in the OKR gain is plotted as a function of the *vestibular weight*, determined as the ratio between the aVOR and aVOR +OKR values at week 0. The *vestibular weight* therefore represents the frequency-dependent relative influence of the vestibular signal on gaze stabilization. *Figure 5D* shows that at week 8, the change in the OKR occurred at the frequencies for which the *vestibular weight* is dominant (>50%), and that the increase in OKR was positively correlated with the *vestibular weight* among frequencies (slope of the

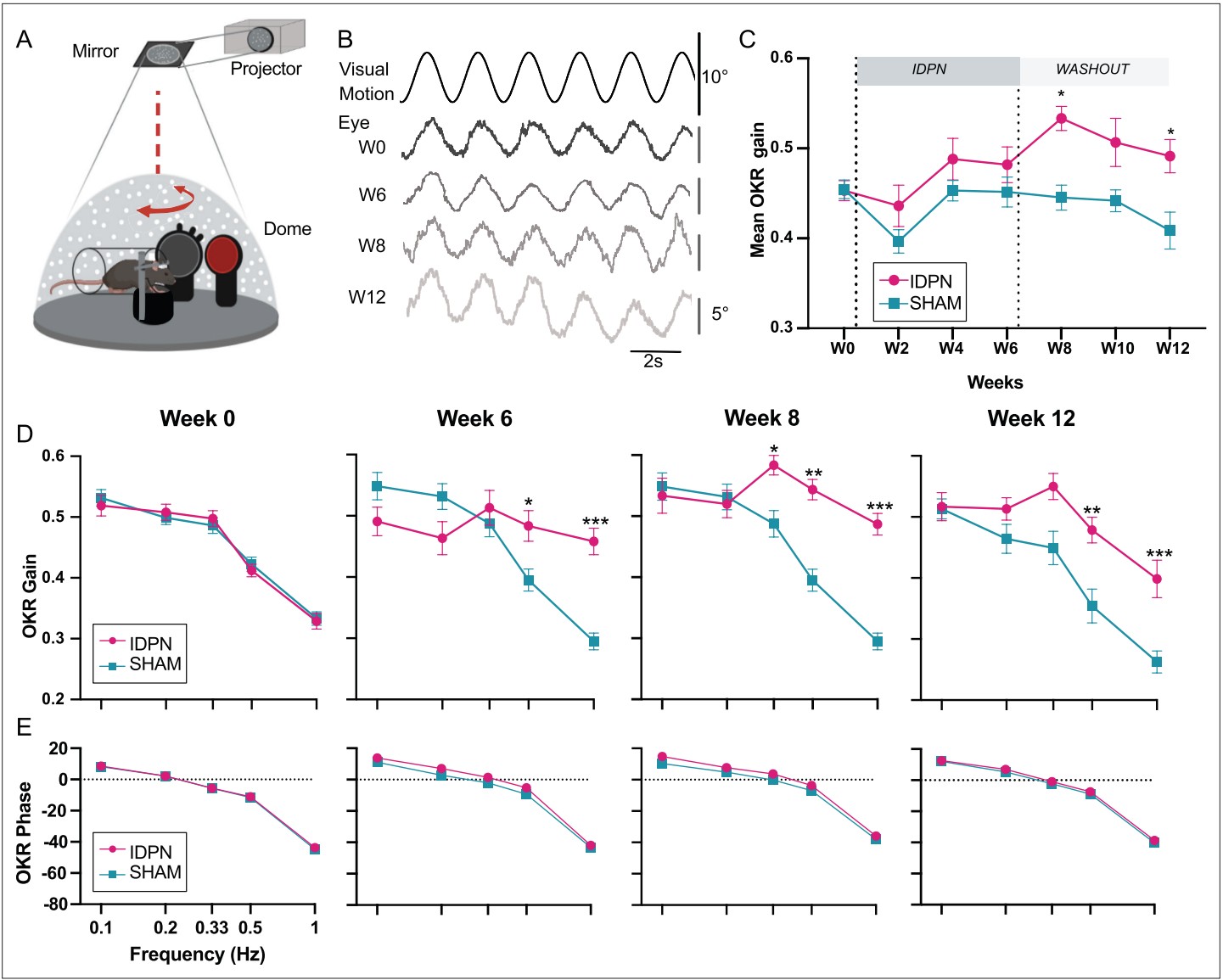

**Figure 4.** Effects of subchronic IDPN treatment on the OKR. (**A**) Illustration of the optokinetic reflex (OKR) set-up. (**B**) Example of raw traces of an OKR recorded in response to stimulation at 0.5 Hz at a peak velocity of 10°/s before (**W0**), after 4 weeks of IDPN treatment (**W4**), 2 weeks (**W8**) and 6 weeks of washout (**W12**). All traces are from the same individual. (**C**) Mean OKR gain of IDPN (n=12) and SHAM (n=12) mice (repeated measures ANOVA). (**D**) OKR gain and (**E**) phase for IDPN (n=12) and SHAM (n=12) for each frequency at W0, W6, W8 and W12 (repeated measures ANOVA). (*p<0.05; **p<0.01; ***p<0.001). Error bars represent ± SEM.

The online version of this article includes the following figure supplement(s) for figure 4:

**Figure supplement 1.** Effect of the IDPN on optokinetic reflex amplitude and timing.

regression line: 1.235, r²=0.6299, p<0.0001). Overall, these results suggest that the increase in OKR gain observed at high frequencies could correspond to a 'visual substitution' occurring primarily in the range where the vestibular inputs are normally dominating gaze stabilization.

## Visuo-vestibular interactions following alteration of vestibular inputs

To investigate whether OKR increase did constitute a 'visual substitution' that maintained optimal gaze stabilization at light despite vestibular loss, we investigated how IDPN-treated mice integrated vestibular and visual inputs. To this end, a session of combined visual and vestibular stimulation (aVOR in light, here referred as *Combined Gaze Response* or CGR condition) was performed at W6 and at W12 (n=19 IDPN; n=12 SHAM). A model (see Materials and methods) was implemented to predict

**Table 2.** Statistics table of the OKR gain for the IDPN-treated group.

| | W0 | W2 | W4 | W6 | W8 | W10 | W12 | |
|---|---|---|---|---|---|---|---|---|
| | | ns | ns | ns | *** | ns | ns | W0 |
| | ns | | ns | ns | *** | ** | ns | W2 |
| | ns | ns | | ns | ns | ns | ns | W4 |
| | ns | ns | ns | | * | ns | ns | W6 |
| | *** | *** | ns | * | | ns | ns | W8 |
| | ns | ** | ns | ns | ns | | ns | W10 |
| | ns | ns | ns | ns | ns | ns | | W12 |

the theoretical CGR (visuo-vestibular) gain and phase from aVOR and OKR (unimodal) measured values. The model was first optimized to correctly predict the average SHAM CGR gains and phases (*Figure 6A*) from their individual aVOR and OKR responses, and then applied to the individual IDPN data. The predicted and observed CGR gain (Model factor) were compared for each treatment group (ANOVA Group x Model, $F_{(1,29)}$ = 12.236, p=0.0013): as expected, model predictions for SHAM well-matched experimental data (Post-hoc Newman-keuls, SHAM MODEL vs SHAM CGR, p=0.41). When applied to IDPN mice the model predicted that their CGR gain should

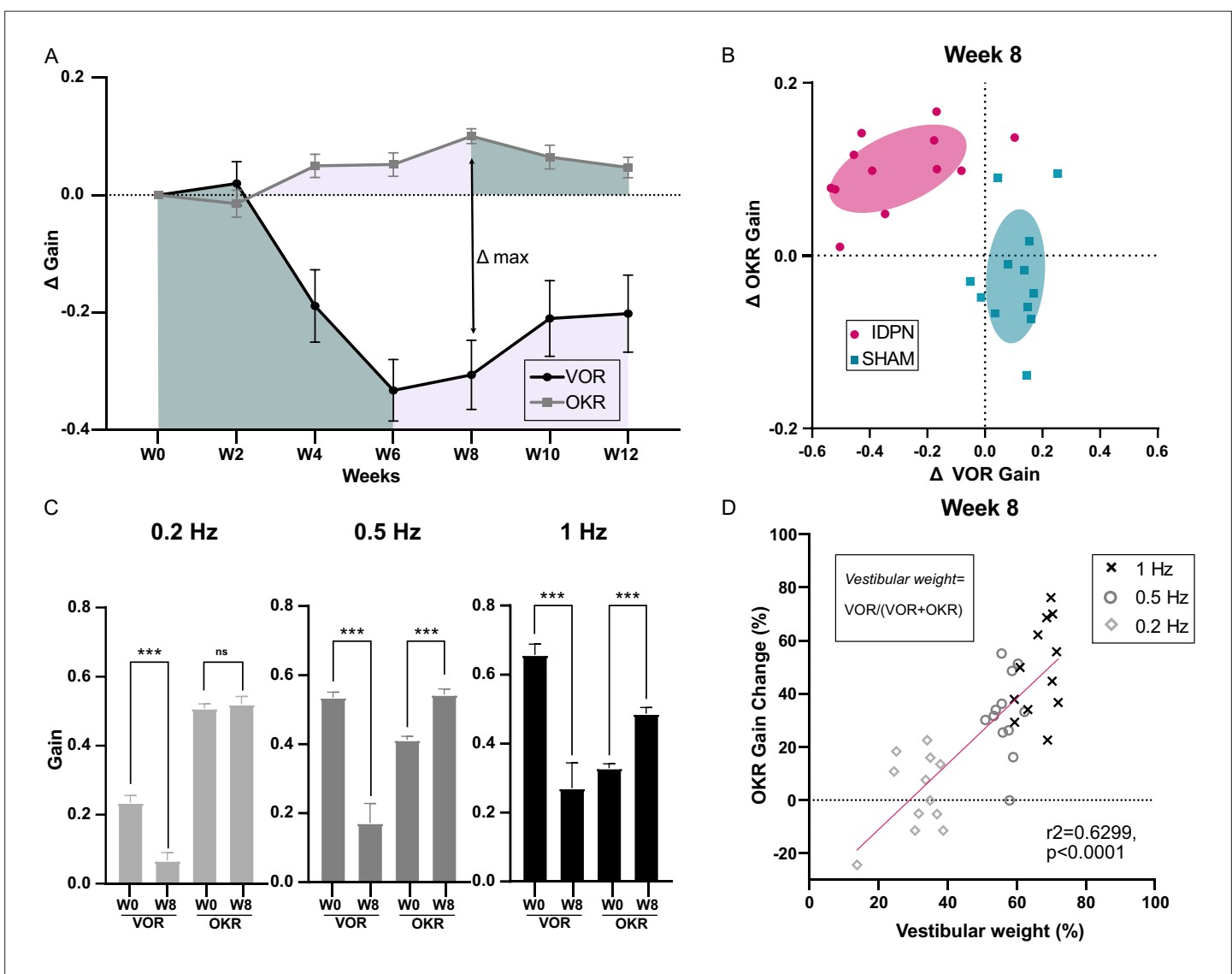

**Figure 5.** Comparison of the IDPN treatment on OKR and aVOR. (**A**) Mean Δ aVOR and Δ Gain OKR for 0.2, 0.5, and 1 Hz for IDPN (n=12) and SHAM (n=12) mice. (**B**) Individual ΔOKR gains as a function of individual ΔaVOR gains. The 50% confidence interval of each group is represented in the shaded areas. (**C**) aVOR and OKR gains of IDPN mice (n=12) at W0 and W8 for frequencies of 0.2, 0.5, and 1 Hz (repeated measures ANOVA). (**D**) Percentage of the individual vestibular weight (inset), as a function of the percentage of the individual OKR gains change for IDPN (n=12). The linear regression corresponds to all values (n=36). (*p<0.05; **p<0.01; ***p<0.001). Error bars represent ± SEM.

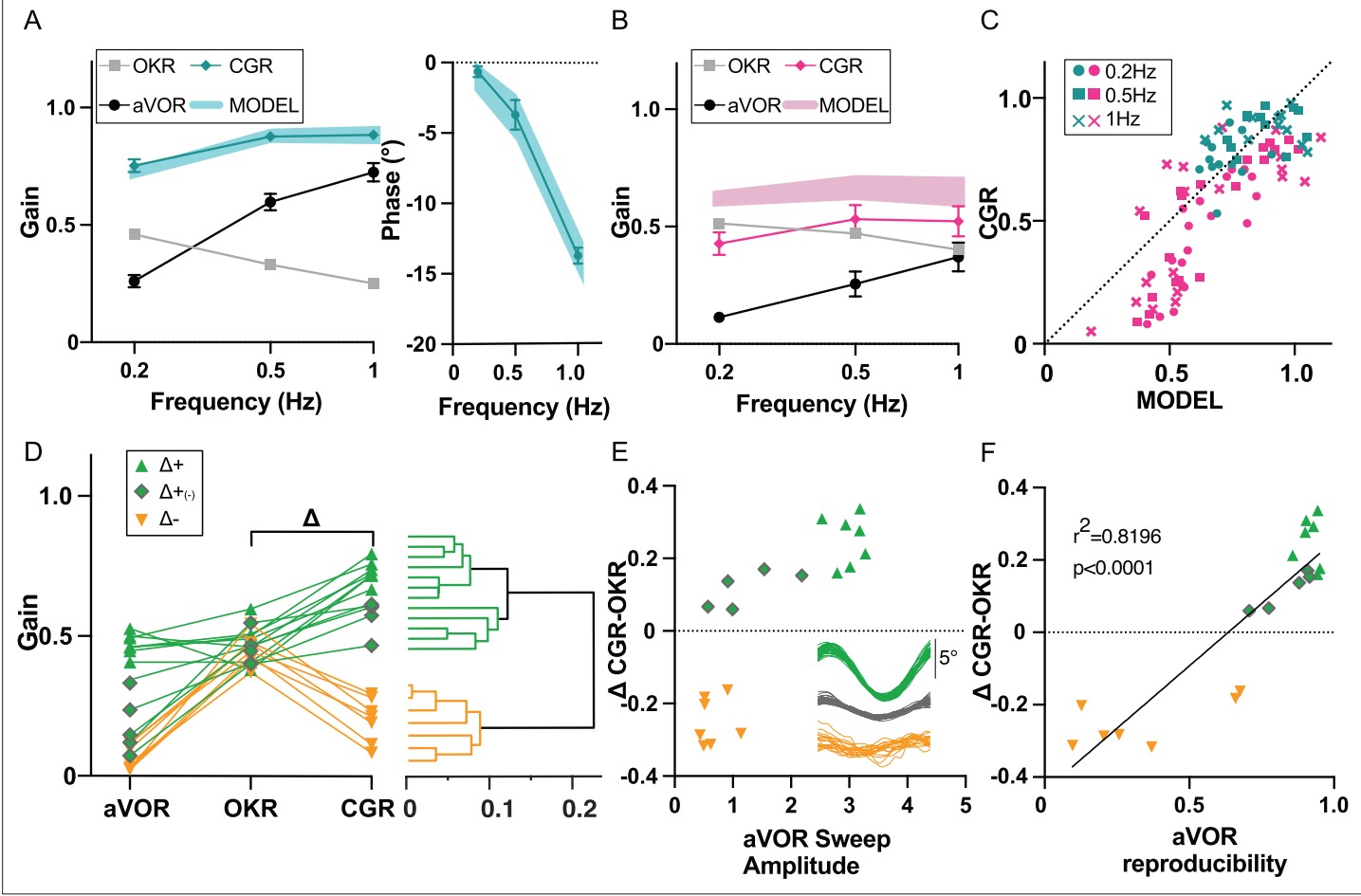

**Figure 6.** Visuo-Vestibular interactions following IDPN treatment. (**A**) aVOR, OKR, CGR data (CGR) and predicted CGR (MODEL) gains (left panel), and CGR phase (data and model, right panel), for the SHAM mice (n=12) at W6 and W12. (**B**) aVOR, OKR, CGR data (CGR) and predicted CGR (MODEL) gains for the IDPN mice (n=19) at W6 and W12. (**C**) Comparison of the predicted versus measured CGR for all frequencies tested. The dotted line at 45° represents a perfect match between prediction and data (optimal CGR). (**D**) aVOR, OKR, and CGR values of the IDPN mice (n=19) (left panel). Clustering analysis (right panel) distinguish two groups based on Delta (CGR-OKR), and a subgroup based on VOR. The horizontal coordinate of each cluster (vertical lines) represents the distance between two connected clusters. (**E**) Δ(CGR-OKR) as a function of the VOR Sweep Amplitude of the Δ+ (green triangle), Δ+(-) (green diamonds) and Δ- (orange) IDPN mice. The inset panel represents typical raw cycles representative of the three subgroups. (**F**) Delta (CGR-OKR) as a function of the VOR reproducibility of the Δ+ (green triangle), Δ+(-) (green diamonds), and Δ- (orange) IDPN mice. Regression line correspond to all values (n=19).

The online version of this article includes the following source data for figure 6:

**Source data 1.** Visuo-Vestibular interactions following IDPN treatment.

be lower compared to SHAM (shaded blue and red areas on *Figure 6A and B*, Post-hoc Newman-keuls, SHAM MODEL vs IDPN MODEL, p=0.00176), suggesting that even an optimal combination of unimodal responses could not restore completely normal CGR in IDPN-treated mice. The CGR gains observed for IDPN-treated mice actually revealed how their visuo-vestibular responses were not only reduced compared to SHAM (compare solid blue and red lines in *Figure 6A–B*, Post-hoc Newman-keuls, SHAM CGR vs IDPN CGR, p=0.00055), but were even inferior to the prediction of the model (compare solid red line and shaded red area in *Figure 6B*, Post-hoc Newman-keuls, IDPN MODEL vs IDPN CGR, p=0.00041). This suggests that the IDPN-treated mice sub-optimally combined their residual/compensated vestibular and visual reflexes to stabilize gaze. *Figure 6C* illustrates the relationship between the measured CGR gain and the prediction of the model for all tested frequencies. For SHAM animals, CGR gain values were positioned along the unity line that represents a close match between predicted and measured CGR (mean ± SEM: SHAM MODEL 0.8255±0.053; SHAM CGR 0.8375±0.042). For IDPN-treated mice, the responses formed two subgroups that did

not correspond to different frequencies (all frequencies found in either subgroup). The first subgroup was intermingled with the SHAM responses, with measured CGR values slightly lower than predicted values (MODEL 0.7457±0.028; CGR 0.6644±0.052). The second subgroup represented mice for which the measured CGR significantly underperformed the predicted CGR (MODEL 0.4685±0.022; CGR 0.2005±0.01). To investigate the difference between the two subgroups, the VOR, OKR, and CGR values were plotted for the n=19 IDPN mice (*Figure 6D*). A hierarchical cluster analysis (see methods) suggested the presence of two main clusters that differ from the clusters observed considering only the aVOR gain (see dendrogram in *Figure 6D*). The first cluster (Δ+, n=12; in green in *Figure 6D, E and F*) comprised individuals with CGR values higher than OKR (positive values on Δ=CGR OKR), that is with gaze stabilization at light better in the presence of vestibular stimulation. The second cluster (Δ-, n=7; depicted in orange) is composed of individuals with low CGR (<0.4). IDPN mice from the Δ- cluster have degraded responses during bimodal visuo-vestibular stimulation compared to unimodal OKR stimulation (negative value of Δ indicates CGR responses lower than OKR responses). Notably, this degraded CGR is not solely a consequence of a low VOR gain, as some Δ+mice identified by the second differentiation of the clustering analyses (depicted with grey diamond symbols in *Figure 6D*, and indicated as Δ+(-)) also showed low unimodal VOR gain responses.

We reasoned that the worsening of the gaze responses in the bimodal condition compared to the OKR-only condition could be a consequence of a poorly reliable, noisy vestibular signal. Based on a cycle-to-cycle sweep analysis, two additional features of the VOR responses were quantified (see methods): the mean amplitude of the raw sweep during VOR stimulation and the VOR response reproducibility between sweeps. The mean VOR raw sweep amplitude could not statistically differentiate Δ- and Δ+(-) mice (*Figure 6E*, orange triangle vs green diamonds p=0.0857), suggesting that vestibular signal of comparable amplitude improved or deteriorated the CGR response in the Δ+(-) and Δ- mice, respectively. On the other hand, the VOR reproducibility clearly discriminated Δ- and Δ+(-) mice (*Figure 6F*, orange triangle vs green diamonds, p=0.0001). There was a strong correlation between the inter-sweep reproducibility and the capacity of the mice to stabilize gaze in presence of vestibular inputs at light. This result suggests that the incapacity of the Δ- group to compute optokinetic signals in the presence of vestibular stimulation could relate to the poor reliability of the vestibular signal. Overall, these data suggest that in some situations the presence of a degraded vestibular signal of significant amplitude (panel 6E) but poorly reliable (panel 6 F) can be detrimental to properly stabilize gaze in presence of visual and vestibular inputs.

## Discussion

### IDPN as a model of partial and transitory loss of vestibular function

IDPN has long been established as an ototoxic compound targeting vestibular HC in the different vestibular endorgans of rats (*Llorens et al., 1993*; *Llorens et al., 1994b*; *Llorens and Demêmes, 1994a*) and later in guinea-pig, frogs, and mice (*Soler-Martín et al., 2007*; *Greguske et al., 2019*). Its ototoxic vestibular effects were used as a tool to study extraocular muscle development (*Brueckner et al., 1999*) or more recently to induce permanent vestibular loss in mice (*Yang et al., 2019*; *Zeng et al., 2020*) and study hair cell regeneration (*Sayyid et al., 2019*). The sub-chronic, reversible protocol used in our study was validated previously in rats (*Sedó-Cabezón et al., 2015*; *Martins-Lopes et al., 2019*; *Maroto et al., 2021*) and mice (*Boadas-Vaello et al., 2017*; *Greguske et al., 2019*) with postural/locomotor quantification of vestibular loss. We for the first time demonstrate that the subchronic protocol leads to a progressive and partly reversible loss of vestibulo-ocular reflexes. These previous studies further demonstrated some of the cellular mechanisms associated with the progressive loss of postural control: ototoxic effects lead to the early dismantlement of calyceal junction, followed by synaptic uncoupling, both of which were shown to be reversible, while continuation of the IDPN treatment would lead to hair cell extrusion and long-term, permanent lesion. HC loss was also demonstrated to occur in a central to peripheral order within vestibular epithelia, and in crista to utricule to saccule order (see *Sedó-Cabezón et al., 2015*). *Maroto et al., 2021* convincingly demonstrated that type I HC show greater sensitivity than type II HC to IDPN subchronic exposure. Given these data, a primary goal of the present study was to try to correlate the loss of VOR functions to organ-specific, zone-specific, and cell-type-specific effects.

Loss and substantial recovery of the vestibular function quantified by VOR measures were found to be correlated with the number of type I HC in both canals and otoliths. A parallel was previously established between HC integrity and VOR following ototoxic protocols in mice (*Cassel et al., 2019*; *Yang et al., 2019*; *Zeng et al., 2020*), but these were established at the population level. Here, we for the first time specifically correlated the loss of HC and loss of VOR function both at the individual level and in an organ-specific way. aVOR and OCR tests demonstrated a parallel decrease in canal- and otolith-dependent functions, respectively. The 6-week-long treatment was followed by a 6-week-washout period, allowing a significant but partial recovery of the aVOR, and complete recovery of the OCR. The individual correlations between these functional tests and the number of HC counted in the ampullae and maculae of the mice were found to be particularly significant, both during the trough (W6) of VOR function and after the recovery period (W12; *Figure 3*). We, however, did not find any difference between the HC loss in the central vs. peripheral zones of the organs, and can therefore not conclude on any putatively differential implication of these areas in the VORs. Effects of the IDPN treatment on the aVOR was evidenced by a gain decrease starting W4, later associated with a significant phase lead at W6 (*Figure 1—figure supplement 1A and B*). The dynamic of these changes might be related to the amount of HC progressively affected by the treatment. The modifications affected all frequencies; however, responses at the lowest tested frequency (0.2 Hz) tended to be less affected and to recover first. These results could suggest that type I HC are crucial element to encode both the amplitude and timing of the VOR, particularly for the more dynamic stimulations. How IDPN progressively impairs the encoding of information by the HC, and how population-coding influence both VOR features in the natural range of head-movements should be the focus of future dedicated studies.

Overall, these organ-specific structure-function correlations confirm that the vestibulo-ocular reflex can serve as a proxy indicating the status of the vestibular endorgans. It further validates the use of the aVOR and OCR as relevant tests reflecting HC integrity at the level of the ampullae and maculae, respectively.

## Differential alteration of type I versus type II hair cells

The subchronic treatment affected type I HC more than type II HC, in accordance with previous reports (*Llorens et al., 1993*; *Maroto et al., 2021*). Susceptibility to IDPN was also reported to vary between different mouse strains (*Boadas-Vaello et al., 2017*; *Wilkerson et al., 2018*). Interestingly, we used a different strain of mice than the one used in a previous subchronic experiment (*Greguske et al., 2019*) and have not observed any type II HC loss either, nor differences between males and females (data not shown).

Hence, IDPN treatment induces a loss of cell markers specific to type I HC, with no effects on type II markers. Two types of markers were used to identify type I HC. Spp1 targeted a protein located on the neck of the hair cell (*McInturff et al., 2018*), while CASPR1 is located at the calyceal junction of the afferent terminal, and has been proven to be necessary for the functionality of the synapse between the HC and its connected afferent (*Sousa et al., 2009*). Although it is not possible to compare for one individual the number of cells before and after treatment, the number of cells labelled for either of these proteins decreased significantly compared to SHAM group, whose numbers are consistent between W6 and W12. The loss of CASPR1 marker induced by IDPN sub-chronic treatment has been linked to HC detachment from the calyx terminal and loss of vestibulo-spinal reflexes, and both CASPR1 expression and vestibular function recover during washout (*Sedó-Cabezón et al., 2015*; *Greguske et al., 2019*; *Maroto et al., 2021*). Only after a longer IDPN exposure does HC extrusion occur, associated with persistent functional deficits. In theory, the loss of vestibular function observed in both canal- and otolith-dependent VOR at W6 could be linked to loss of type I HC. In adult rodents, however, the regeneration of hair cells leads to the formation of cells with type II HC features (*González-Garrido et al., 2021*), so the recovery of type I cellular markers at W12 is likely not a result of cell regeneration. Also, the apparent dramatic loss of CASPR1 and Spp1 HC (90%) at W6 was in contrast with the much smaller and not significant loss in Myo7a cells (29%), supporting the conclusion that the sensory cells persisted in the epithelium despite CASPR1 loss, and that the recovery in the number of type I HC observed at W12 was more likely the result of molecular repair, not cell regeneration. In addition, we observed a similar loss of Spp1 marker at W6, normally located in the neck of the hair cell. As such the loss of vestibular function could rather be attributed to a global disorganization of the synapse and defective hair cell function, associated with a transitory

large decrease of the Spp1 and CASPR1 proteins and a smaller decrease in Myo7a. In accordance with this hypothesis, the number of Spp1 and CASPR1 positive cells after the washout period at W12 was not significantly different from the SHAM group and correlated to the recovery of the vestibular function observed at W12 for the IDPN mice (*Figure 3C2*). The altered expression of proteins in type I HC induced by the sub-chronic treatment of IDPN seems to be reversible in most individuals. One possibility is that definitive extrusion of type I HC occurred in the most susceptible individuals that did not show recovery after the end of treatment. This in fact was the case in the 2 individuals tested at W12 with immuno-histochemistry: CASPR1 mean value (± S.E.M) of 18.5±5.5 compared to 34.5±4.359 for IDPN with recovery and 41.75±1.215 for SHAM. It was previously reported in the pigeon that type I HC and calyx afferents take longer (12 weeks) to regenerate following aminoglycoside toxicity, while type II and boutons endings regenerated in a week (*Zakir and Dickman, 2006*). Similarly, recovered innervation for type I HC was delayed compared to type II HC in mice (*Kim et al., 2022*). A longer recovery period could be investigated in the case of the severely-affected mice to confirm the absence of recovery in the longer term.

## Evidence for the role of type I HC in the vestibulo-ocular pathways

To our knowledge, the direct implication of type I and type II HC in the VOR was never directly tested. Type I and type II HC differ by many features, including their anatomical location within the epithelium, electrophysiological properties, morphology and innervation by afferents (*Eatock and Songer, 2011*). Seemingly, irregular afferents have been described as predominantly innervating the central area and striolar zones of the ampullae and maculae with calyx and dimorphic synaptic contacts, while regular afferents make dimorphic and buttons contacts predominantly within the peripheral zones of cristae and extrastriolar zones of maculae (*Goldberg, 2000*; *Eatock and Songer, 2011*; *Contini et al., 2022*). Afferents with bouton terminals that contact only type II HC have regular discharge, while afferents with calyx terminals that contact only type I HC have irregular discharge. However, most regular and irregular afferents are fed by both type I and type II HC (*Goldberg et al., 1992*). Functionally, regular afferents recorded in the monkey show a lower detection threshold than irregular afferents (*Sadeghi et al., 2007*), while irregular afferents showing higher gains and phase leads (mouse: *Lasker et al., 2008*; *Cullen, 2019*) would be better optimized for encoding high dynamic stimuli (*Cullen, 2019*). Central VOR neurons receive a mixture of regular and irregular afferent inputs (*Goldberg et al., 1987*; *Boyle et al., 1992*; *Goldberg, 2000*), with irregular afferents constituting ~1/3 of their excitatory drive (*Goldberg et al., 1987*; *Boyle et al., 1991*).

While the role of type I and type II HC in the VOR remains to be determined, previous studies have emphasized the importance of the regular afferents and regular central vestibular neurons in the vestibulo-ocular pathway. Functional ablation targeting irregular afferents suggested that VOR might function normally with only intermediate and regular afferent inputs (*Minor and Goldberg, 1991*). Functionally identified VOR neurons (i.e. PVP) were recently demonstrated to have heterogeneous discharge variability (high or low), with the most regular units particularly well-suited to faithfully encode the compensatory eye movements generated during natural stimulation (*Mackrous et al., 2020*).

Based on this literature, it would be tempting to infer that only the most regular and tonic elements of the entire vestibular pathway are responsible for the VOR. However, the correspondence between hair cell properties and afferents/central neurons properties is only partial, such that the two phasic/irregular and tonic/regular channels for head motion signals are constituted of both types I and type II HC (*Baird et al., 1988*; *Goldberg et al., 1990* p.90) and afferents (irregular and regular) (*Goldberg, 2000*; *Eatock and Songer, 2011*; *Beraneck and Straka, 2011*). *Carey et al., 1996* previously reported a better correlation of VOR recovery with type I than with type II HC following ototoxic lesions in the chick. Overall, the result of ototoxic studies, including the present one, demonstrate a fundamental role of type I HC in the encoding of vestibular signals that drive the vestibulo-ocular reflexes, even in the relatively low range (*Carriot et al., 2017*) of head movements tested. It has also been recently shown that both type I and type II HC actually contribute to otolithic vestibular evoked potential responses (i.e. vestibulo-spinal pathway), previously described as mostly type I-specific (*Sayyid et al., 2019*). Overall, our results are compatible with the hypothesis of a convergence of heterogeneous peripheral neural elements at the level of central vestibular nuclei, where intrinsic properties of central vestibular neurons (*Straka et al.,*

*2005*; *Beraneck and Idoux, 2012*) supplemented by network properties (*Beraneck et al., 2007*; *Pfanzelt et al., 2008*; *Beraneck and Straka, 2011*) would differentially integrate vestibular signals further processed in the different functional pathways (*Sadeghi and Beraneck, 2020*; *Mackrous et al., 2020*).

## Visuo-vestibular interactions after IDPN treatment

OKR plasticity following vestibular loss concerned gain, and not phase, modifications for frequencies >0.33 Hz. Previous studies have reported an increase in OKR gain following a permanent vestibular lesion for a non-specific range of frequencies (*Shinder et al., 2005*; *Faulstich et al., 2006*; *Nelson et al., 2017*). One key to understand the frequency-specific OKR plasticity could be the physiological dominance of vestibular inputs in the gaze stabilization process at high frequencies (*Faulstich et al., 2004*). The visual inputs could be reweighted and potentiated specifically in the range where the vestibular loss has the most impact on gaze stabilization (*Figure 5D*).

The VOR and OKR work synergistically to stabilize gaze by compensating for head and visual surround movements, respectively (*França de Barros et al., 2020*). If the reflexive eye movements are not perfectly compensatory, an error signal (e.g retinal slip) is produced that drives adaptation of the VOR (*Boyden et al., 2004*; *Dean and Porrill, 2014*; *Shin et al., 2014*) and OKR (*Glasauer, 2007*; *Kodama and du Lac, 2016*). In an effort to understand the integration of VOR and OKR (*Holland et al., 2020*) recently proposed that OKR would account for the retinal slip not compensated by VOR. Within that framework, the changes in the OKR could happen preferentially as a function of the vestibular weight, that is in the range of movements where the loss of vestibular inputs generates the largest retinal slip. While present data are evidence for frequency-specific adaptation, how this relates to alteration in vestibular inputs and/or visual feedback signals remains to be determined.

The increase in OKR gain could be seen as a substitution for the decrease in the VOR. However, even after VOR recovery, the optimal integration of OKR and VOR (as predicted by our MODEL) led to a degraded combined gaze response (CGR) with respect to SHAM mice. Even worse, the measured CGR was inferior to the predicted CGR (*Figure 6B*). This observation suggests that the integration of visual and vestibular signals during CGR is more than a mere summation of the gain and phase of the two unimodal reflexes, but is also affected by other factors. In fact, a subset of the IDPN individuals showed severely degraded bimodal responses, so that their combination of VOR and OKR is not only sub-optimal but also less effective than the unimodal (OKR) reflex. A sweep-based analysis suggested that in these individuals, the VOR unimodal responses were not reliable, i.e the low reproducibility indicates that the same stimuli are unfaithfully encoded into eye movements. We interpret this low reproducibility as a sign of noisiness within the VOR pathway that would not only preclude the optimal integration of both sensory signals, but also disturb the use of retinal information to stabilize the gaze. According to statistical optimality theories, as the Maximum Likelihood principle, the decrease in VOR reliability should lead to a reduction in its relative sensory weight. This filtering-out of a degraded signal has been shown during visuo-vestibular integration for monkey heading perception (*Fetsch et al., 2010*), or for human object discrimination using visual and haptic senses (*Ernst and Banks, 2002*).Our results suggest that the gaze stabilizing system is not able to optimally adapt to the degradation by filtering-out vestibular signals during combined visuo-vestibular combination.

The integration of visual and vestibular inputs occurs in several structures, including in the cerebellar flocculi (*Jang et al., 2020*) and brainstem (*Carcaud et al., 2017*; see review *De Zeeuw et al., 2021*). In the case of vestibular loss, most of the defects are expected to concern the central vestibular neurons involved in the VOR, which also integrate visual inputs (ES, or eye-sensitive neurons; *Beraneck and Cullen, 2007*). After unilateral neurectomy, vestibular neurons in the Deiters nuclei responded to higher frequencies during visual stimulation (Cat: *Zennou-Azogui et al., 1994*), a change compatible with the hypothesis that OKR gain increase partly takes place in the vestibular nuclei. It was shown that inhibitory floccular inputs (part of the OKR indirect pathway) and excitatory vestibular inputs often colocalized on the dendrites of central vestibular neurons. One possibility is therefore that the massive disorganization of vestibular periphery inputs led in the long term to synaptic and intrinsic changes at the level of central vestibular neurons (*Beraneck and Idoux, 2012*; *Carcaud et al., 2017*), thus impairing their capacity to optimally integrate both signals.

## Conclusions

Balance dysfunction occurs frequently in aging people (85% prevalence above 80; *Agrawal et al., 2013*) but also in younger people (prevalence of 35% in the 40-year-old). While animal models often use permanent vestibular lesions (*Simon et al., 2020*), many diseases in fact consist in a gradual, transient and/or partial loss of vestibular function. For instance Ménière's disease, which represents ~9% of all vestibular pathologies in the adult and occurs in the <60 years old, is characterized by recurrent episodes with brief (<24 hr) fluctuating symptoms and otherwise normal vestibular function before long-term deterioration arises (*Lopez-Escamez et al., 2015*). Transitory vestibular symptoms are also commonly reported in vestibular neuritis where, although symptoms tend to rapidly disappear due to vestibular compensation, vestibular function can recuperate up to a year after the initial loss (*Welgampola et al., 2019*). Gradual and partial vestibular loss is also encountered as a side-effect of some ototoxic anti-cancer treatments (prevalence in treated patients ~40%; cisplatin; *Paken et al., 2016*; *Prayuenyong et al., 2018*). Here, we took advantage of a subchronic ototoxic protocol to determine how animals adapt to partial and transitory loss of vestibular hair cells. We show that the loss, and then the partial recovery of the VOR, is correlated with the integrity of the type I HC, demonstrating for the first time their essential role in the VOR, whether of canalar or otolithic origin. We show that sensory (visual) substitution would theoretically compensate for vestibular loss, but that injured mice have suboptimal responses when combining visual and vestibular information. Finally, we show that this impairment in multisensory integration would be linked to the loss of 'reliability' of the vestibular signal, degraded by ototoxicity. Overall, these results suggest that transitory peripheral infraction have long term consequences, and that the capacity of central vestibular structures to cope (vestibular compensation; *Cullen et al., 2010*; *Beraneck and Idoux, 2012*; *Lacour et al., 2016*) with the sensorineural loss might critically depend on the integrity of the neural elements involved. Future studies should aim at obtaining information about the degradation of signal transmission following IDPN treatment and characterize the amount of peripheral population-coding necessary to preserve optimal vestibular function.

# Materials and methods

**Key resources table**

| Reagent type (species) or resource | Designation | Source or reference | Identifiers | Additional information |
|---|---|---|---|---|
| Antibody | Anti-Myosin 740 VIIa (rabbit polyclonal) | Proteus Biosciences | Cat#:25–6790 | 1/400 |
| Antibody | Anti-rabbit IgG H+L (donkey polyclonal) | Jackson Immuno Research | RRID:AB_2340616 | 1/500 |
| Antibody | Anti-contactin-associated protein (mouse monoclonal) | Neuromab | Cat#:75–001 | 1/400 |
| Antibody | Anti-mouse IgG H+L (donkey polyclonal) | Life Technologies | RRID:AB_141607 | 1/500 |
| Antibody | Anti-calretinin (guinea-pig polyclonal) | Synaptic Systems | Cat#: 214 104 | 1/500 |
| Antibody | Anti-guinea-pig IgG H+L (donkey polyclonal) | Jackson Immuno Research | RRID:AB_2340476 | 1/500 |
| Antibody | Anti-osteopontin (goat polyclonal) | R&D Systems | Cat#:AF808 | 1/400 |
| Antibody | Anti-goat IgG H+L (donkey polyclonal) | Invitrogen | RRID:AB_2535853 | 1/500 |
| Software | Spike2 | Cambridge Electronic Design | RRID:SCR_000903 | |
| Software | Prism | GraphPad | RRID:SCR_002798 | |
| Software | ImageJ | National Institutes of Health | RRID:SCR_003070 | |

## Headpost implantation surgery and animal care

Surgical procedures, postoperative care, device fixation and animal surveillance during the protocol were performed as described previously in *França de Barros et al., 2019* Briefly, 6-weeks-old mice anaesthetized with isoflurane gas had their heads shaved with small clippers. Then, lidocaine hydrochloride (2%; 2 mg/kg) was injected locally before a longitudinal incision of 1.5 cm was made to

expose the skull. Just anterior to the lambda landmark, a small custom-built headpost (3x3 × 5 mm; poly lactic acid) was cemented (C&B Metabond; Parkell Inc, Edgewood, NY) and laterally covered with resin (Heraeus) for protection. Animals were fully recovered 30 min after the end of the surgery, yet buprenorphine (0.05 mg/kg) was provided for postoperative analgesia and they were closely monitored for the following 48 hr.

## Subchronic ototoxic exposure

Twenty-eight mice were treated with 3,3′-iminodiproprionitrile (IDPN, Sigma Aldrich, 317306, 90%) for 6 weeks (IDPN group) at 30 mM concentration of IDPN in their drinking water at least 72 hr after the surgery. After 6 weeks of treatment, a washout period of 6 weeks followed. Previous experiments (*Greguske et al., 2021*) had demonstrated that at these concentrations, ototoxic lesions produced by IDPN are largely reversible. The SHAM group was tested the exact same way but was not exposed to IDPN in their drinking water. Video-oculography tests were performed before the beginning of the treatment and once every two weeks until week 12 (W12), for a total of 7 sessions of tests for each mouse.

## Experimental groups

3 different batches of mice were used, each composed of both IDPN-treated mice and SHAM. The mice of the first one (n=19) were subjected only to vestibular stimulations, to test the effect of IDPN on the canal-related or otolith-related VOR responses during the 12 weeks of protocol. The second group (n=24), designed to compare the dynamic of VOR and OKR changes was tested with angular VOR stimulations, otolithic test, optokinetic (visual) stimulations, and combined vestibular and visual stimulation (VOR in light; referred to as Combined Gaze Response or CGR). The third group (n=11) designed to correlate ototoxic effects of IDPN on vestibular hair cells (HC) and VOR function was tested for canal-related or otolith-related VOR responses, before treatment and after 6 weeks of treatment and immediately used for immunohistochemical assessment of the vestibular sensory epithelia (see below). A subset of the second group (n=12 out of 24) was also used for immunohistochemical assessment at the end of the 12 weeks of protocol.

## Video oculography recording sessions

The recording procedure was similar to the one presented in *Carcaud et al., 2017 França de Barros et al., 2020*. The animals were head-fixed in a custom build Plexiglas tube, and their head was oriented in a 30° nose down position to align the horizontal canals to the yaw plane. Their left eye movements were recorded using a non-invasive video oculography system (ETL –200 Iscan, acquisition rate 120 Hz, *Stahl et al., 2000*). Eye, head, and virtual drum (OKR stimulation) position signals were digitally recorded (CED mk3 power 1401, acquisition rate 1 kHz) with Spike 2 software. Signals were analysed offline in Matlab (Matlab, the MathWorks, Natick, MA, USA; RRID, SCR:001622) programming environment. The restraining apparatus was fixed on a rotation platform on top of an extended rig with a servo-controlled motor. Single VOR or OKR recording sessions lasted no longer than 45 min in total.

## Vestibular stimulations and analysis

All vestibular-specific tests were performed in a temperature-controlled room with all sources of light turned off except for computer screens. The turntable was further surrounded with a closed box to isolate the animal from remaining light, with a final luminance inside the box <0.02 lux. Myosis was induced with topical 2% pilocarpine applied 10 min before experimentation.

*Vestibulo ocular reflex in dark (VORd) tests* were performed in the dark with the mouse surrounded by an opaque black dome (*Figure 1A*). Sinusoidal angular rotations around the vertical axis were used to record the horizontal angular vestibulo-ocular reflex (aVOR), at different frequencies: 0.2, 0.5, 0.8; 1 and 2 Hz at a peak velocity of 30°/s.

*Angular vestibulo-ocular reflex* analysis was similar to the one described in *Carcaud et al., 2017*. Segments of data with saccades were excluded from VOR slow-phase analysis. For horizontal sinusoidal rotations, at least 10 cycles were analyzed for each frequency. VOR gain and phase were determined by the least-squares optimization of the equation:

$$EH_v\left(t\right) = g \cdot \left\{ \left[ HH_v \cdot \left(t - t_d\right) \right] + C^{te} \right\}$$

where EHv(t) is eye horizontal velocity, g (gain) is a constant value, HHv (t) is head horizontal velocity, td is the dynamic lag time (in msec) of the eye movement with respect to the head movement, and C^te is an offset. The td was used to calculate the corresponding phase (φ°) of eye velocity relative to head velocity. The Variance-Accounted-For (VAF) of each fit was computed as

$$1 - \left[ \frac{var\left(est - EHv\right)}{var\left(EHv\right)} \right]$$

where *var* represents variance, *est* represents the modeled eye velocity, and *EHv* represents the actual eye horizontal velocity. VAF values for VOR measures were typically between 0.70–1 (>95% of recordings), where a VAF of 1 indicates a perfect fit to the data. For IDPN-treated mice, abnormally low (<0.10) values of gain associated with VAF <0.5 were nevertheless included in the gain statistical analysis but specifically reported in grey (*Figure 2*, *Figure 1—figure supplement 1*). Corresponding phase values were not included in the statistical analysis of the aVOR phase.

*Static Ocular Counter-roll reflex tests* were performed as described in *Simon et al., 2021* by measuring the vertical eye movement generated in response to different roll angles. The table was moved from left to right in incremental steps of 10° (from 0 to 40°), with static periods of at least 10 s between oscillations (*Figure 1D*) to record the stabilized eye elevation and declination. The OCR gain corresponds to the slope of the linear regression of the vertical eye angle *vs.* the head tilt angles (*Oommen and Stahl, 2008*).

*Off vertical axis rotation (OVAR) tests* were performed with the vestibular turntable tilted with a 17° off-axis angle following the methodology described in *Beraneck et al., 2012 Idoux et al., 2018*. 50°/s continuous stimulations were performed in a counter-clockwise and then clockwise direction. OVAR generates a continuous change in the head tilt angle, compensated through a maculo-ocular reflex (MOR) by the generation of a horizontal nystagmus compensating for the table constant rotation. This oculomotor response was quantified following the methodology described in *Idoux et al., 2018*. First, quick-phases were identified and removed. During rotations, the velocity of horizontal slow phases is modulated (modulation, μ) around a constant bias (β). Both parameters (μ and β) were calculated from the sinusoidal fit of eye horizontal slow-phase velocity using the least-squares optimization of the equation:

$$SP\left(t\right) = \beta + \mu \cdot sin\left[2\pi \cdot f_0 \cdot \left(t + t_d\right)\right]$$

where SP(t) is slow-phase velocity, β is the steady-state bias slow phase velocity, μ is the modulation of eye velocity, f₀ is the frequency of table rotation, t_d is the dynamic lag time (in msec) of the eye movement with respect to the head movement. The bias (Maculo-ocular reflex Bias) is reported here as the main index of the MOR response (*Hess and Dieringer, 1990*; *Beraneck et al., 2012*).

## Optokinetic reflex tests and analysis

Horizontal optokinetic stimulations were performed as previously described in *França de Barros et al., 2020*. The mice were placed under a semi-opaque plastic dome and all sources of light were turned off. The projected stimulation consisted of a randomly distributed white dots pattern on a black background image (250000 white dots, max width 0.075°). The optokinetic sinusoidal stimulations were tested at 0.1, 0.2, 0.33, 0.5 and 1 Hz at a peak velocity of 10°/s. The gain and phase were obtained by the same least-squares optimization method described above for the aVOR. To prevent putative cross effects between visual and vestibular stimulations, VOR and OKR test sessions were performed on separate days.

## Combined visual and vestibular stimulations

Combined visual and vestibular stimulations measuring the combined gaze reflex (CGR) consisted of horizontal vestibular stimulations while projecting the fixed dotted pattern used for OKR on the surrounding dome. Horizontal angular sinusoidal rotations were performed at frequencies of 0.2, 0.5 and 1 Hz with a peak velocity of 30°/s. The CGR gain and phase quantifications were performed following the same methodology as for aVOR. To avoid interference, these were performed on

n=8 SHAM and n=12 IDPN after the end of the VOR and OKR test sessions on W12. To test the effects at W6, a specific group of mice (n=4 SHAM and n=7 IDPN) was tested at W0 and W6.

A simple model has been developed to predict for each individual and for three stimulation frequencies (0.2, 0.5, and 1 Hz) the theoretical CGR response gain and phase were given the same two parameters observed for their unimodal VOR and OKR responses. The experimental VOR and OKR gain and phase are used to compute the sinusoid representing the two unimodal responses. The gain and phase of the CGR sinusoid resulting from adding the VOR and OKR sinusoids are quantified by measuring its peak-to-peak amplitude and by identifying the time-lag that maximizes its cross-correlation with the sinusoidal stimulus (note that the sum of two sinusoids with the same period, even if with different phases, is always a sinusoid). Independently of the mouse, the unimodal VOR and OKR sinusoids are shifted in time before their summation, in order to improve, on average, the fit between predicted and observed CGR responses for SHAM mice. The two optimal shift parameters, one for VOR and one for OKR, obtained for each frequency, are then used also for all IDPN-treated mice.

To identify subpopulations of IDPN-treated mice, we computed an agglomerative hierarchical cluster tree on a dataset composed of individual aVOR gain and Δ (calculated as CGR-OKR gain). The classification method used unweighted average Euclidean distance between clusters.

To describe the mice VOR responses without any assumption about their sinusoidal nature the two following characteristics have been quantified: the reproducibility of the ocular response to the vestibular stimulation and the amplitude of the raw eye movements generated by the vestibular stimulation. These analyses were performed on the slow phases horizontal eye trajectories, *EHp(t)*, recorded during each sweep of stimulation.

The reproducibility index, *Rep*, was obtained by computing the matrix of correlation coefficient, $R_{i,j}$, between each couple (*i,j*), of the *N* selected sweeps, and then computing the mean of all above-diagonal elements of the matrix as reported in the following equation (this choice aims at considering only once each $R_{i,j}$, since $R_{j,i}=R_{i,j}$ and to exclude diagonal elements, $R_{i,i} = 1$).

$$Rep = \frac{\sum_{i=1}^{N-1} \sum_{j=i+1}^{N} R_{i,j}}{N\left(N-1\right)/2}$$

The amplitude, *Amp*, of the raw eye movements was quantified by computing for each sweep how much on average, during a stimulation cycle, the eye moved horizontally away from its average position ($\sigma_{EHp}$). Before averaging this parameter over all the sweep, it had to be squared. Finally, to have a parameter value in degrees, the root mean squared of the mean was computed.

$$Amp = \sqrt{\frac{\sum_{i=1}^{N} \sigma_{EHp}^{2}}{N}}$$

## Immunolabelling of the vestibular HC

Two groups of mice (n=11 at W6 and n=12 at W12) were used to perform immunofluorescence analysis on hair cells in the vestibular endorgans. Mice were anaesthetized with an overdose of intraperitoneal injection of ketamine hydrochloride (10%)/xylazine (5%) and decapitated. The histology was done following the protocol established by *Lysakowski et al., 2011*, as described previously (*Maroto et al., 2021*). The vestibular epithelia (one horizontal canal and one utricule) were dissected and fixed for 1 h in a 4% solution of paraformaldehyde (PFA). PFA was washed twice with phosphate-buffered saline (PBS) and the samples were then embedded in a cryoprotective solution (34.5% glycerol, 30% ethylene glycol, 20% PBS, 15.5% distilled water) to be stored at –20°. Before the immunochemistry, samples were put at room temperature and rinsed twice in PBS. While under slow agitation, the samples were incubated twice, first for 1 hr with a 4% Triton X-100 (Sigma Aldrich) in PBS to permeabilise and a second time for 1 hr in 0.5% Triton X-100 1% fish gelatin (CAS #9000-70-8, Sigma-Aldrich) in PBS to block. The incubation with the primary antibodies was then performed in 0.1 M Triton X-100, 1% fish gelatin in PBS at 4° for 24 hr. After rinsing, the second antibodies were incubated in the same conditions. The 2nd antibodies were rinsed and the vestibular epithelia were mounted on slides with fluoromount (F4680, Sigma-Aldrich) and were visualised with a confocal microscope Zeiss LSM880 (with an objective of 63 x NA:1.4). To properly analyse the whole vestibular epithelium, Z-stacks of 0.5 μm were obtained and observed with ImageJ (National Institute of Mental Health, Bethesda,

Malyland, USA). The primary antibodies used were rabbit anti-Myosin VIIa (Myo7a) (Proteus Biosciences, #25–6790), mouse anti-contactin-associated protein (CASPR1) (Neuromab #75–001), guinea pig anti calretinin (Synaptic Systems #214–104) and goat anti-osteopontin (Spp1) (R&D Systems #AF808). Their respective secondary antibodies were Dylight 405 donkey anti-rabbit igG H+L (Jackson Immuno Research #711-475-152), Alexa Fluor 488 donkey anti-mouse IgG H+L (Life Technologies #A21202), Alexa Fluor donkey anti-guinea-pig IgG H+L (Jackson ImmunoResearch #706-605-148) and Alexa Fluor 555 donkey anti-goat IgG H+L (Invitrogen #A21432).

HC counts were obtained from square areas (67.5x67.5 µm) of the central and peripheral parts of two vestibular organs (horizontal semi-circular canal and utricle) in the W6 group and the W12 group. The areas for counting were obtained from the same location within the epithelium in all animals. The global number of HC was assessed with the cytoplasmic labelling of the anti-Myo7a antibody (*Hasson et al., 1997*). Type I HC were labelled using two different antibodies: anti-Spp1 in the neck of the type I HC (*McInturff et al., 2018*) and the presence of anti-CASPR1 in the calyceal junctions of the calyx (*Sousa et al., 2009*). Type II HC were distinguished by the colocalization of Myo7a and calretinin (*Maroto et al., 2021*).

## Statistical analysis

For both OKR and VOR stimulations, the effect of the protocol on the gain and phase were statistically tested by performing repeated measures ANOVA. The *treatment* (SHAM or IDPN) was considered as between individual independent factor with the *Weeks* (W0, W2, W4, W6, W8, W10, W12) and the *Frequencies* (0.1, 0.2, 0.33, 0.5 and 1 Hz for the OKR; 0.2, 0.5, 0.8, 1 and 2 Hz for the VOR) were considered as within individual independent factors. The main effects of those factors and their interactions were tested and reported. In the case of the OCR and OVAR, mix-model ANOVA was used with only *Weeks* considered as within factors. For the CGR, measured at W6 and W12, mix-model ANOVA was used with the *Frequencies* and *Model* (Measured or Model CGR) used as within factors. For the comparison between the OKR and VOR Δ gain, a repeated measures ANOVA was applied on the Δ gain (Wx-W0) with the *stimulation modality* (OKR or VOR) as between individual independent factor and the *Weeks* as within individual independent factor. For the correlation between OCR and VOR, as well as the measured and theoretical CGR gain a linear regression model was fitted.

The effect of IDPN exposure on the HC count was reported with a Kruskal Wallis test. For all analyses the significance threshold was set at $P<0.05$ and Newman Keuls post hoc test was performed if the main effect or an interaction was found significant.

## Acknowledgements

This study contributes to the IdEx Université de Paris ANR-18-IDEX-0001. This work has benefited from the support and expertise of the animal facility of BioMedTech Facilities at Université Paris Cité (Institut National de la Santé et de la Recherche Médicale Unité S36/Unité Mixte de Service 2009). The confocal microscopy studies were performed at the Centres Científics i Tecnològics de la Universitat de Barcelona (CciTUB). This work is supported by the Centre National d'Etudes Spatiales, the Centre National de la Recherche Scientifique, and the Université Paris Cité. MB and LS received funding from the Agence Nationale de la Recherche (ANR-20-CE37-0016 INVEST). MB, FS & DF received funding from the ERANET NEURON Program VELOSO (ANR-20-NEUR-0005). AP & JL received funding from the ERANET NEURON Program VELOSO (grant PCI2020-120681-2 from MCIN/AEI/10.13039/501100011033 and NextGenerationEU/PRTR).

## Additional information

### Funding

| Funder | Grant reference number | Author |
| --- | --- | --- |
| Agence Nationale de la Recherche | ANR-20-CE37-0016 | Louise Schenberg Mathieu Beraneck |

| Funder | Grant reference number | Author |
|---|---|---|
| Agence Nationale de la Recherche | ERANET NEURON ANR-20-NEUR-0005 | François Simon<br>Desdemona Fricker<br>Mathieu Beraneck |
| Agencia Estatal de Investigación | ERANET NEURON PCI2020-120681-2 | Aïda Palou<br>Jordi Llorens |
| Centre National d'Etudes Spatiales | DAR CNES 4500072566 | Michele Tagliabue<br>Mathieu Beraneck |

The funders had no role in study design, data collection and interpretation, or the decision to submit the work for publication.

## Author contributions

Louise Schenberg, Conceptualization, Data curation, Software, Formal analysis, Supervision, Investigation, Visualization, Methodology, Writing - original draft, Writing – review and editing; Aïda Palou, Data curation, Formal analysis, Investigation, Visualization, Methodology, Writing – review and editing; François Simon, Methodology, Writing – review and editing; Tess Bonnard, Investigation, Visualization; Charles-Elliot Barton, Investigation; Desdemona Fricker, Supervision, Funding acquisition, Writing – review and editing; Michele Tagliabue, Software, Formal analysis, Supervision, Methodology, Writing - original draft, Writing – review and editing; Jordi Llorens, Conceptualization, Resources, Formal analysis, Supervision, Funding acquisition, Validation, Methodology, Project administration, Writing – review and editing; Mathieu Beraneck, Conceptualization, Resources, Data curation, Software, Formal analysis, Supervision, Funding acquisition, Validation, Visualization, Methodology, Writing - original draft, Project administration, Writing – review and editing

## Author ORCIDs

Louise Schenberg  http://orcid.org/0009-0004-1298-8228
Desdemona Fricker  http://orcid.org/0000-0001-7328-9480
Michele Tagliabue  http://orcid.org/0000-0002-4905-788X
Jordi Llorens  http://orcid.org/0000-0002-3894-9401
Mathieu Beraneck  http://orcid.org/0000-0003-2722-0532

## Ethics

A total of 56 C57BL/6 J mice in an equal partition of males and females were used for the protocol. Animals were used in accordance with the European Communities Council Directive 2010/63/EU. All efforts were made to minimize suffering and reduce the number of animals included in this study. All procedures were approved by the ethical committee for animal research of the Université Paris Cité.

Reviewer #1 (Public Review): https://doi.org/10.7554/eLife.88819.3.sa1
Reviewer #2 (Public Review): https://doi.org/10.7554/eLife.88819.3.sa2
Author Response https://doi.org/10.7554/eLife.88819.3.sa3

# Additional files

## Supplementary files
- MDAR checklist
- Source data 1. Source data for *Figures 4 and 5*.

## Data availability

All data generated or analysed during this study are included in the manuscript and supporting file; source data files have been provided for all figures.

The following dataset was generated:

| Author(s) | Year | Dataset title | Dataset URL | Database and Identifier |
|---|---|---|---|---|
| Beraneck M, Schenberg L | 2023 | Schenberg et al. 2023 | https://doi.org/10.17632/byrsp7tpdb.1 | Mendeley Data, 10.17632/byrsp7tpdb.1 |

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
