## [Editor Report · eLife assessment]

This paper provides a **fundamental** expansion of vestibular compensation into transient and partial dysfunction, as well as insights into the adaptation of visual reflexes in this process. The conclusions are **convincingly** supported with paired histological and behavioral measurements, which are additionally modeled for further interpretation. This work would be of interest to neuroscientists working in multisensory integration and recovery mechanisms.

---

## [Referee Report · Reviewer #1 (Public Review)]

To further understand the plasticity of vestibular compensation, Schenberg et al. sought to characterize the response of the vestibular system to short-term and partial impairment using gaze stabilization behaviors. A transient ototoxic protocol affected type I hair cells and produced gain changes in the vestibulo-ocular reflex and optokinetic response. Interestingly, decreases in vestibular function occurred in coordination with an increase in ocular reflex gain at frequencies where vestibular information is more highly weighted over visual. Moreover, computational approaches revealed unexpected detriment from low reproducibility on combined gaze responses. These results inform the current understanding of visual-vestibular integration especially in the face of dysfunction.

Strengths

The manuscript takes advantage of VOR measurements that can be activated by targeted organs, are used in many species including clinically, and indicate additional adverse effects of vestibular dysfunction.

The authors use a variety of experimental procedures and analysis methods to verify results and consider individual performance effects on the population data.

The conclusions are well-justified by current data and supported by previous research and theories of visuo-vestibular function and plasticity.

---

## [Referee Report · Reviewer #2 (Public Review)]

This is a very nice study showing how partial loss of vestibular function leads to long term alterations in behavioural responses of mice. Specifically, the authors show that VOR involving both canal and otolith afferents are strongly attenuated following treatment and partially recover. The main result is that loss of VOR is partially "compensated" by increased OKR in treated animals. Finally, the authors show that treatment primarily affects type I hair cells as opposed to type II hair cells. Overall, these results have important implications for our understanding of how the VOR Is generated using input from both type I and type II hair cells.

The major strength of the study lies in the use of partial inactivation of hair cells to look at the effects on behaviors such as VOR and OKR. Some weaknesses stem from the fact that the effects of inactivation are highly variable across specimens and that there is no recovery of behavioral function.

---

## [Author Response]

The following is the authors’ response to the original reviews.

We would like to thank the editors and reviewers for their overall positive assessment of this work. We have carefully revised the manuscript and implemented near all reviewers’ public and confidential recommendations. We believe these modifications have strengthened the manuscript and hope it will further convince the editors and reviewers.

We below provide a point-by-point response to the reviewers’ comments.

**Reviewer #1 (Public Review):**
To further understand the plasticity of vestibular compensation, Schenberg et al. sought to characterize the response of the vestibular system to short-term and partial impairment using gaze stabilization behaviors. A transient ototoxic protocol affected type I hair cells and produced gain changes in the vestibulo-ocular reflex and optokinetic response. Interestingly, decreases in vestibular function occurred in coordination with an increase in ocular reflex gain at frequencies where vestibular information is more highly weighted over visual. Moreover, computational approaches revealed unexpected detriment from low reproducibility on combined gaze responses. These results inform the current understanding of visual-vestibular integration especially in the face of dysfunction.StrengthsThe manuscript takes advantage of VOR measurements which can be activated by targeted organs, are used in many species including clinically, and indicate additional adverse effects of vestibular dysfunction. The authors use a variety of experimental procedures and analysis methods to verify results and consider individual performance effects on the population data. The conclusions are well-justified by current data and supported by previous research and theories of visuo-vestibular function and plasticity.

The authors thank reviewer 1 for emphasizing these positive aspects of the work.

WeaknessesThe manuscript describes the methodology as inducing reversible changes (lines 44, 67,) but the data shows a reversible effect only in hair cell histology (Fig 3A-B) not in function as demonstrated by the persistent aVOR gain reduction in week 12 (Fig 1C) and increase of OKR gain in weeks 6-12 (Fig 4C/D).

Rodents exposed to IDPN in the drinking water show from complete to null reversibility of the function loss depending on the IDPN concentration and duration of exposure, and the relationship between exposure and effect varies as a function of species, strain and sex of the exposed animals (Llorens and Rodríguez-Farré, Neurotoxicol. Teratol., 1997; Seoane et al., J. Comp. Neurol. 2001; Sedó-Cabezón et al., Dis. Model. Mech., 2015; Greguske et al., Arch. Toxicol., 2019). In addition, there is individual variability. The concentration of IDPN and time of exposure used in this study were selected to result in a loss followed by complete reversion but, as noted by the referee, the reversion was complete on Hair cells, while the gaze stabilizing reflexes show differential degrees of recovery depending on the functional tests (complete recovery on OCR; partial on aVOR and OKR). These make the IDPN subchronic protocol an interesting methodology to study the long term consequences of partial/reversible inner ear impairment. To be more accurate in the description of the reversibility, we have now introduced the following changes:

Lines 43: Subchronic exposure to IDPN in drinking water at low doses allowed for progressive ototoxicity, leading to a partial and largely reversible loss of function.

Lines 67-68: We demonstrate that despite the significant recovery in their vestibulo-ocular reflexes, the visuo-vestibular integration remains notably impaired in some IDPN-treated mice

Lines 578: Previous experiments (Greguske et al., 2021) had demonstrated that at these concentrations, ototoxic lesions produced by IDPN are largely reversible.

Reviewer 1: The manuscript begins with the mention of fluctuating vestibular function clinically, but does not connect this to any specific pathologies nor does it relate its conclusions back to this motivation.

Thank you. We have now added a conclusion (lines 525-552) to discuss the results in a clinical perspective.

Reviewer 1: The conclusions of frequency-specific changes in OKR would be stronger if frequency-specific aVOR effects were demonstrated similar to Figure 4D.

We have presented the frequency-selective effects in Figure 1 supplement and related text; changes observed in aVOR are mostly (see below) comparable for all frequencies >0.2Hz. However, we have edited the text to better highlight when the IDPN differentially affect aVOR tested at different frequencies (see lines 97-99).

**Reviewer #2 (Public Review):**
This is a very nice study showing how partial loss of vestibular function leads to long term alterations in behavioural responses of mice. Specifically, the authors show that VOR involving both canal and otolith afferents are strongly attenuated following treatment and partially recover. The main result is that loss of VOR is partially "compensated" by increased OKR in treated animals. Finally, the authors show that treatment primarily affects type I hair cells as opposed to type II. Overall, these results have potentially important implications for our understanding of how the VOR Is generated using input from both type I and type II hair cells. As detailed below however, more controls as well as analyses are needed.

The authors thank reviewer 2 for positive evaluation regarding the potential implication of the work.

Major points:Reviewer 2: The authors analyze both canal and otolith contributions to the VOR which is great. There is however an asymmetry in the way that the results are presented in Figure 1. Please correct this and show time series of fixations for control and at W6 and W12. Moreover, the authors are plotting table and eye position traces in Fig. 1B but, based on the methods, gains are computed based on velocity. So please show eye velocity traces instead. Also, what was the goodness of fit of the model to the trace at W6? If lower than 0.5 then I think that it is misleading to show such a trace since there does not seem to be a significant VOR.

Figure 1 was modified as suggested. Panel B now shows velocity traces, and goodness of fit is reported in figure legend. Panel E now shows raw OCR traces at W0, W6, W12.

Reviewer 2: This is important to show that the loss is partial as opposed to total. It seems to me that the treatment was not effective at all for aVOR for at least some animals. What happens if these are not included in the analysis?

The reviewer is correct, there is indeed variability in the alteration observed during the treatment, as previously described and expected from previous experiments (Llorens and Rodríguez-Farré, Neurotoxicol. Teratol., 1997; Seoane et al., J. Comp. Neurol. 2001; SedóCabezón et al., Dis. Model. Mech., 2015; Greguske et al., Arch. Toxicol., 2019). It was actually one of the goal of the study to compare hair cell loss and VOR responses in individuals. The individual aVOR gain and phase responses during the IDPN treatment are all presented in Figure 1 supplement. aVOR was reduced in all animals, although 2/21 only showed a decrease of less than 10% of their initial gain at W6. If these were excluded from the analysis, the statistical differences between the 2 groups would be reinforced.

Reviewer 2: Figure 2A shows a parallel time course for gains of aVOR and OCR at the population level. Is this also seen at the individual level?

Yes, this is seen in individuals. This result is presented in Figure 2C and 2D which illustrate the similar effect of IDPN on aVOR and OCR responses at week 6 and week 12 at the individual level (each symbol represents an individual mouse). The plotted delta gain of both aVOR and OCR represents the relative loss of vestibular function for each individual mouse at W6 and W12, respectively.

Reviewer 2: Figure 3: please show individual datapoints in all conditions.

Figure 3 was modified to show individual datapoints in all conditions (see Figure 3 A2, A3, C2 and C3).

Reviewer 2: Figure 4: The authors show both gain and phase for OKR. Why not show gain and phase for aVOR and OCR in Figure 1. I realize that phase is shown in sup Figures but it is important to show in main figures. The authors show a significant increase in phase lead for aVOR but no further mention is made of this in the discussion. Moreover, how are the authors dealing with the fact that, as gain gets smaller, the error on the phase will increase. Specifically, what happens when the grey datapoints are not included?

As pointed by the reviewer, we have included all aVOR phase results in Figure 1 supplement and also stated it in the main text (lines 100-102). There is however no phase calculated for the OCR which is a static test, as better illustrated in new Figure 1E.Error in phase calculations increases as gain gets smaller. To take this into account, the phase corresponding to the grey points (VAF<0,5; corresponding to Gains<0.10) were not included in the statistical analysis of the aVOR phase. This point is now made clearer in methods lines 639-640.

Reviewer 2: Discussion: As mentioned above, the authors should discuss the mechanisms and implications of the observed phase lead following treatment. Moreover, recent literature showing that VN neurons that make the primary contribution to the VOR (i.e., PVP neurons) tend to show more regular resting discharges than other classes (i.e., EH cells), and that such regularity is needed for the VOR should be discussed (Mackrous et al. 2020 eLife). Specifically, how are type I and type II hair cells related to discharge regularity by central neurons in VN?

We have now added discussion regarding mechanisms and implications of the phase changes in lines 363-371. The authors thank reviewer 1 for pointing at the Mackrous et al. 2020 eLife paper which is now included in the updated discussion. The relations between type I and type II and discharge regularity in afferents and central VN is further discussed 442-449.

Below we provide answers to specific recommendations for the authors.

**Reviewer #1 (Recommendations For The Authors):**
Reviewer 1: Were hair cells counted for the whole organ? what was the control for epithelial size differences?

The effect of the treatment on hair cells was estimated by counting numbers of cells in square area of the central and peripheral parts of the sensory epithelia. The text has been modified to better describe the method, lines 748-751.

Reviewer 1: The title of the article leads readers to expect more emphasis on hair cell changes, while the content of the manuscript is more functional and encompassing the visual and vestibular systems.

We have retained the original tittle.

Reviewer 1: Please provide acronym definitions before they are used. Examples: HC (line 63), W6 etc (line 82-83)

Done as suggested on lines 63, 82 and 107.

Reviewer 1: Please describe the ages of animals used in the study.

The animals used in the study were 6 weeks old at the beginning of the protocol and 20 weeks old at the end. The text has been modified accordingly, line 564.

Reviewer 1: Consider changing "until" to "through" when describing time ranges (initially line 88), as the following time mentioned is included in the statement. E.g., line 216-217 sounds as if gain was insignificantly different at W12.

Done as suggested, lines 88 and 219.

Reviewer 1: Line 162: lower case for "immunostaining".

Done, line 164.

Reviewer 1: Consider regrouping or renumbering panels of Figure 3 for more clarity.

Panels in Figure 3 were regrouped as suggested, with first the canal-related data in panels A-B followed by the utricule-related data in panels C-D.

Reviewer 1: Lines 222-223: reword as gain increased not frequency.

Thank you, the text has been reworded, line 224-225.

Reviewer 1: It is unclear if the two subgroups revealed in CGR analysis (line 288) are relevant to the two groups described in VOR responses (line 137-138). Please clarify if these are the same mice or distinct clusters.

The two subgroups found in the CGR analysis differ from the clusters revealed by the decrease of the aVOR gain; the text has been modified lines 300-301.

Reviewer 1: Consider adding that irregular afferents + calyces are relevant specifically to type I HCs (lines 411-426).

The text has been modified to clarify the contacts between the two types of vestibular afferents and hair cells, lines 431-435.

Reviewer 1: Line 434: clarify which "scheme" given context before and after this phrase

In order to clarify this part of the discussion, the text has been modified and this term no longer appears.

Reviewer 1: Please indicate the time gap from surgery to treatment.

The time gap from the surgery to treatment, at least 72h, has been updated in the methods, lines 575.

Reviewer 1: Line 619-620: It is unclear if VOR and OKR sessions were randomized in order or if the authors have considered training or adaptive effects from the initial test session.

VOR and OKR sessions were performed on different days to limit cross effects, lines 639-640.

Reviewer 1: Line 688: typo-change ifG to IgG.

modified, line 744.

Reviewer 1: Line 692-693: were hair cells counted for the whole organ? what was the control for epithelial size differences?

The effect of the treatment on hair cells was estimated by counting numbers of cells in square area of the central and peripheral parts of the sensory epithelia. The text has been modified to better explain the method, lines 748-751.

Reviewer 1: Change decimal indicator to be consistent (commas used in lines 732, 759, 776, Figure 6C),

Thank you; modified as suggested.

Reviewer 1: Line 763: "stimulation at 0.5Hz &10{degree sign}/s" is unclear.

The text has been modified, line 817.

Reviewer 1: Line 765: bold (E)

The police format has been updated, line 820.

Reviewer 1: Line 770-771: (A) insert OKR to be "mean delta aVOR and delta OKR gain", (B) plot is OKR as a function of VOR.

Thank you, done as suggested. The text has been modified, line 824. Reviewer 1: Describe Figure 6 delta at initial mention (line 784 instead of 786) Authors: thank you, done as suggested, line 839.

Reviewer 1: It is unclear why the tables are included if never mentioned in the text.

The tables are now mentioned, lines 90 and 218.

Reviewer 1: Figure 1: is the observed gain for Sham group expected value rather than closer to 1?

Yes, as the value reported on Figure 1 is a mean of the values obtained during aVOR test in the dark at frequencies in range 0.2-1Hz (see also Figure 1 Supplement).

Reviewer 1: Figure 2: (A) give enough space to see error bars at W2. Consider making test data more easily distinguishable. (B) is OCR mean or specific stimulation? (C/D) move 1Hz label from title to x-axis label as it does not describe OCR test. Figure 5: (C) consider making color specific to frequency for better distinction on C+D as symbols previously indicated individual data. (D) 1Hz specific to OKR? move to axis label instead of title

The Figures 2 and 5 have been modified according to reviewer 1 suggestions.

Reviewer 1: Figure 6: (A/B) what time point are these, W12?

Those points correspond to W6 and W12, the text has been updated to specify the time points, lines 834 and 835.

**Reviewer #2 (Recommendations For The Authors):**
The authors should perform additional analyses that will help strengthen their results.

We are unsure about the exact implementation of this recommendation. However, we have strengthened our results by following all reviewers’ specific recommendations.